# HiPOOD: Hierarchical Prompt-Aware Zero-Shot Out-of-Distribution Detection

## Abstract

Reliable image recognition systems must not only classify known categories accurately but also detect instances of novel, unseen classes in open-set scenarios. Achieving this in a zero-shot setting, without any training examples—remains a significant challenge. In this paper, we propose a zero-shot out-of-distribution (OOD) detection approach that leverages semantic class hierarchies to enrich each known label with fine-grained subcategory sets, capturing subsumption relationships between classes. To generate these hierarchies, we query a large language model (LLM) with structured prompts, producing semantically coherent candidate subcategories that are subsequently filtered with a lexical ontology to ensure domain alignment. We incorporate the resulting label hierarchy into CLIP's classification pipeline, a pre-trained vision–language model (VLM). This design enables the model to distinguish fine-grained categories within the known classes and to recognize when an input does not fit any known class, effectively identifying it as an unknown object. Notably, our approach operates in a zero-shot manner, requiring no additional training. Experiments on several standard OOD detection benchmarks show that our method achieves competitive performance. Furthermore, by organizing predictions within a semantic hierarchy, the model's outputs become more informative and easier to interpret, including for inputs that it flags as unknown.

## 1 Introduction

A common assumption in many visual recognition systems is the *closed-set* setting, where the training and test data are expected to share the same set of labels Hendrycks & Gimpel (2018). In open-world scenarios, this assumption fails: machine learning models must operate under the assumption that novel, previously unseen inputs may arise at test time. A critical challenge in this setting is to identify and reject inputs from unknown classes, a task known as *out-of-distribution* (OOD) detection. When such inputs are not properly handled, models tend to exhibit overconfident predictions, potentially leading to severe errors or safety risks—particularly in high-stakes domains like autonomous driving or medical diagnostics. Therefore, robust OOD detection is essential to ensure the reliability and trustworthiness of AI systems deployed in the real world.

Traditional approaches to visual OOD detection primarily rely on a model's predicted confidence scores or probability outputs to differentiate in-distribution (ID) samples from unknown classes Hsu et al. (2020); Liu et al. (2020); Huang et al. (2021); Sun et al. (2021); Wang et al. (2021). These methods typically exploit properties such as maximum softmax probability, temperature scaling, input perturbations, distances in feature space (e.g., Mahalanobis distance), or energy-based scores derived from network logits to identify OOD inputs Hendrycks & Gimpel (2018); Liang et al. (2018); Lee et al. (2018); Liu et al. (2020). Despite their success on standardized benchmarks, these approaches generally operate under closed-set assumptions, and thus often remain overly confident when encountering novel inputs, especially when unknown classes are semantically close to known categories. To address these limitations, recent research has begun exploring *open-vocabulary vision-language models* (VLMs) such as CLIP Radford et al. (2021), which have introduced a promising direction for zero-shot OOD detection by effectively leveraging multimodal representations. Capitalizing on this capability, recent works have explored OOD detection using textual prompts to query whether an image matches labels described in natural language. For instance, Maximum Concept Matching (MCM) aligns image features with a broad set of generic textual concepts, significantly enhancing the detection of semantically similar unknown inputs Ming et al. (2022).

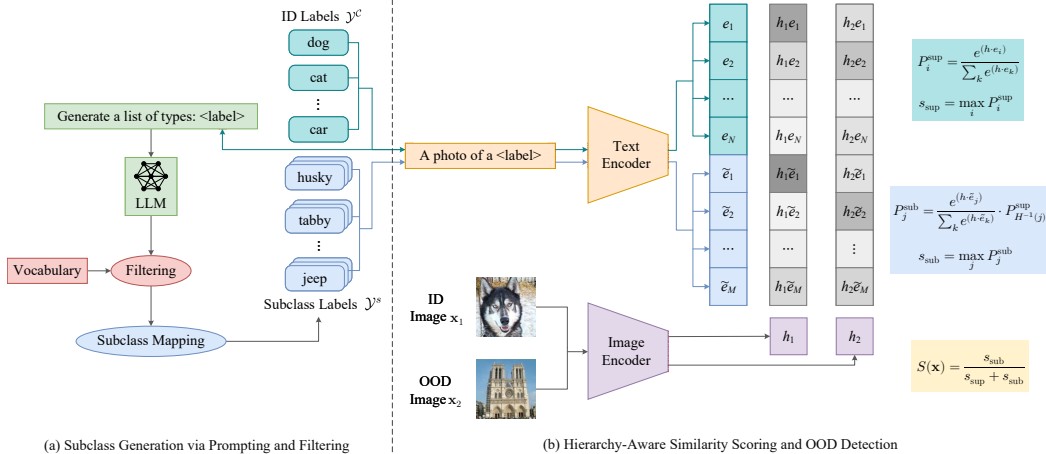

(a) Subclass Generation via Prompting and Filtering  (b) Hierarchy-Aware Similarity Scoring and OOD Detection

Figure 1: Overview of HiPOOD. **(a):** The image encoder maps the input image $\mathbf{x}$ into a normalized embedding $\mathbf{h}$. The text encoder encodes both coarse ID labels $\mathcal{Y}^C$ and fine-grained subclass labels $\mathcal{Y}^S$ into vectors $e$ and $\tilde{e}$. Subclass labels are generated via LLM prompting and filtered through a vocabulary alignment stage. **(b):** At inference, similarities $h \cdot e$ and $h \cdot \tilde{e}$ are computed. Each subclass score is reweighted by the probability of its parent superclass, enabling hierarchy-aware confidence estimation. The HiPOOD score fuses the top coarse-class probability with the maximum reweighted subclass probability to detect OOD samples.

Zero-shot OOD Detection with CLIP (ZOC) generates candidate unknown labels through a trained text captioner and computes confidence based on both known and generated labels Esmaeilpour et al. (2022). Additionally, NegLabel Jiang et al. (2024) expands the OOD detection capability by incorporating a vast collection of negative labels selected from large-scale lexical corpora, while CLIPN Wang et al. (2023) leverages negative prompts to better differentiate ID and OOD inputs. CATEX Liu et al. (2023) goes one step further by learning hierarchical context descriptions from labeled ID data, effectively using supervised, learnable prompts attached to each category to improve OOD performance, but at the cost of additional training on ID images and task-specific context parameters that limit its applicability in strictly training-free or data-scarce deployments. However, the zero-shot methods above (MCM, ZOC, CLIPN, NegLabel) either depend exclusively on predefined ID labels without fully exploiting textual interpretative capabilities (MCM), rely on generated labels that might be ineffective at scale (ZOC), or employ flat negative labels without capturing structured semantic relationships (NegLabel). Therefore, significant potential remains untapped to further enhance zero-shot OOD detection by integrating structured semantic information within VLM-based approaches. Moreover, these approaches often assume that OOD samples are semantically distant from the in-distribution classes, an assumption that may not hold in practice. A more fundamental issue is that even a very large negative label set can never exhaustively cover the space of all possible OOD classes. There will always be unseen unknowns that are not represented by any pre-collected negative label, especially as the open-world visual space is essentially unbounded. Our HiPOOD approach, by contrast, explicitly leverages class hierarchies within the VLM embedding space to refine OOD scores across semantic levels.

In this work, we target settings where label spaces admit a meaningful hierarchy and domain experts are accustomed to hierarchical reasoning—such as biodiversity monitoring (biological taxonomy), medical imaging (clinical ontologies like SNOMED/UMLS Chang & Mostafa (2021); Bodenreider (2004)), retail and e-commerce (product category graphs). We propose to *model hierarchical label structures* to achieve more robust and interpretable OOD detection in zero-shot vision-language settings. Instead of using only the given class names, we automatically organize the labels into a *coarse-to-fine hierarchy* of superclasses and subclasses. By leveraging a hierarchy of semantic prompts, our approach examines an image's alignment with both high-level categories and fine-grained subcategories. The key intuition is that an image in distribution (ID) should yield consistent similarity scores at hierarchical levels - strongly aligned with both a subclass and its associated superclass - while an OOD image will exhibit weaker or inconsistent signals at these levels. We generate this hierarchy automatically, inspired by the CHiLS framework Novack et al. (2023), by using prompt-based queries to a large language model (LLM) to propose candidate subclasses. These candidates are subsequently filtered using CLIP embeddings and a vocabulary to retain only the semantically relevant subclasses, ensuring the hierarchy's quality and coherence. This approach is data-free, relying exclusively on the initial ID class labels and pre-trained models. By integrating

hierarchical semantic consistency into zero-shot OOD detection, our method makes the detection process interpretable: when an image is flagged as OOD, we can inspect which subclass it was closest to and how it deviated from the expected parent class, providing human-understandable insights.

In summary, our contributions are as follows:

- We automatically generate fine-grained subclass labels from the observed ID set via prompt-based queries to an LLM and vocabulary filtering, ensuring predictable and controlled OOD risk without manual annotation.

- We formulate a zero-shot OOD detection score, HiPOOD, which takes advantage of the hierarchy by comparing image-text alignment at coarse and fine levels. This score effectively captures anomalies as images that violate the expected parent-child alignment, allowing the detection of OOD samples with greater sensitivity to semantic nuance.

- HiPOOD provides improved interpretability through explicit hierarchical reasoning, scales efficiently to large label sets, and achieves competitive zero-shot OOD detection without additional training.

Taken together, HiPOOD introduces a new zero-shot OOD detection paradigm based on hierarchical consistency over VLM-derived subclasses, enabling stronger discrimination than existing flat or prompt-based approaches while requiring no supervision or learned prompts.

## 2 BACKGROUND AND NOTATION

Let $\mathcal{X}$ denote the input image space and $\mathcal{Y}^C = \{y_i^C\}_{i=1}^N$ represent the set of in-distribution (ID) class labels, where each $y_i^C$ is a textual label (e.g., *cat*, *dog*, *bird*). The number of ID classes is denoted by $N$. We define $X_{\text{in}} \sim \mathbb{P}_{X_{\text{in}}}$ and $X_{\text{out}} \sim \mathbb{P}_{X_{\text{out}}}$ as random variables drawn from the ID and out-of-distribution (OOD) marginal distributions over $\mathcal{X}$, respectively.

### 2.1 CLIP FOR ZERO-SHOT PREDICTION

CLIP Radford et al. (2021) is a vision-language model trained on roughly 400 million image-text pairs, comprising an image encoder $\mathbf{f}^{\text{img}}$ and a text encoder $\mathbf{f}^{\text{text}}$. For zero-shot classification, CLIP computes embeddings for an input image $\mathbf{x} \sim \mathbb{P}_{X_{\text{in}}}$ and a set of class labels $y_i^C \in \mathcal{Y}^C$. Specifically, given an image $\mathbf{x}$, the model extracts an embedding $\boldsymbol{h} = \mathbf{f}^{\text{img}}(\mathbf{x}) \in \mathbb{R}^D$, where $D$ is the embedding dimension. Simultaneously, each label $y_i^C$ is transformed into a textual embedding $\boldsymbol{e}_i = \mathbf{f}^{\text{text}}(\text{prompt}(y_i^C)) \in \mathbb{R}^D$, using a predefined textual template. The model predicts the class by calculating the cosine similarity between the $\boldsymbol{h}$ and each label embedding $\boldsymbol{e}_i$, assigning the label corresponding to the highest similarity:

$$\hat{y} = \underset{y_i^C \in \mathcal{Y}^C}{\operatorname{argmax}} \left\{ \cos(\boldsymbol{h}, \boldsymbol{e}_i) \right\}. \tag{1}$$

Beyond classification, this mechanism enables diverse CLIP-based tasks such as zero-shot object detection and zero-shot OOD detection, highlighting the flexibility and generalization capacity of CLIP across various application domains Pratt et al. (2023); Santurkar et al. (2022).

### 2.2 PROBLEM FORMULATION

In zero-shot classification with a pre-trained CLIP-like model, the goal is to assign an input image $\mathbf{x} \in \mathcal{X}$ to one of the predefined textual labels $\mathcal{Y}^C = \{y_1^C, \ldots, y_N^C\}$, without requiring any fine-tuning on in-distribution (ID) categories Radford et al. (2021). Each label $y_i^C$ is encoded as a natural language prompt, and classification is performed by comparing the image embedding to the set of label embeddings.

To perform OOD detection under this zero-shot setting, a scoring function $S(\mathbf{x})$ is defined to assess whether a sample belongs to the known ID set. Given a threshold $\gamma$, a sample is considered in-distribution if $S(\mathbf{x}) \geq \gamma$, and out-of-distribution otherwise:

$$G_\gamma(\mathbf{x}) = \text{ID} \ \text{ if } S(\mathbf{x}) \geq \gamma, \ \text{ else OOD}. \tag{2}$$

The challenge lies in designing a score function $S$ that effectively separates ID and OOD samples—assigning higher scores to in-distribution data and lower scores to anomalies—without degrading the model's classification accuracy.

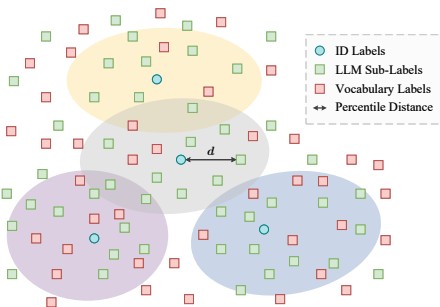

**Figure 2:** Hierarchical label sets construction. Subclass candidates are generated via LLMs, filtered with vocabulary constraints, and selected based on semantic distance to ID labels in CLIP embedding space.

---

**Algorithm 1:** HPSL Construction

1: **Input:** $\mathcal{Y}^C = \{y_i^C\}_{i=1}^N$, $\mathbf{f}^{\text{text}}$, $\{\mathcal{Y}_i^g\}$, $\mathcal{Y}^v$, $M$

2: **Output:** $H : y_i^C \mapsto \mathcal{Y}_{y_i^C}^s$

3: **Encode ID.** For each $y_i^C$: $\boldsymbol{e}_i^C \leftarrow \frac{\mathbf{f}^{\text{text}}(\text{prompt}(y_i^C))}{\|\mathbf{f}^{\text{text}}(\text{prompt}(\cdot))\|_2}$

4: **for** $y_i^C \in \mathcal{Y}^C$ **do**

5:    **Candidates.** $\mathcal{Y}_i^p \leftarrow \mathcal{Y}_{y_i^C}^g \cup \mathcal{Y}^v$

6:    **Similarity score.** For $y_j^p \in \mathcal{Y}_{y_i^C}^p$:

$$\boldsymbol{e}_j^p = \mathbf{f}^{\text{text}}(\text{prompt}(y)/\|\mathbf{f}^{\text{text}}\text{prompt}(\cdot)\|_2$$

$$d_j = \text{percentile}_\eta\big(\{1 - \cos(\boldsymbol{e}_j^p, \boldsymbol{e}_k^C)\}_{k=1}^N\big)$$

7:    **Selection.** $\mathcal{Y}_{y_i^C}^S \leftarrow \text{topk}(\{d_j\}, M)$

   $H(y_i^C) \leftarrow \mathcal{Y}_{y_i^C}^S$

8: **end for**

---

With HiPOOD, we use a hierarchical label set of class names to compute a semantic consistency score across coarse and fine levels. By aligning image features not only with known classes but also with their structured subclasses, HiPOOD improves the model's sensitivity to semantic mismatches while maintaining interpretability and zero-shot performance.

## 3 PROPOSED METHOD

We propose HIPOOD, a hierarchical OOD detection method that builds subsumption label sets over the ID classes and scores each test image with a pre-trained CLIP vision–language model. As illustrated in Figure 1, starting from the ID categories we automatically generate, for each parent class, a set of semantically related subclasses under a subsumption relation. At inference, we encode the image with CLIP and compute similarities to all label embeddings (parents and their subclasses). These scores are then normalized by *two separate softmax distributions*: a coarse-level softmax over parents and a fine-level softmax *per parent* over its subclasses; the fine probabilities are subsequently reweighted by their parent probability. The final OOD score is obtained as a *ratio of two maxima from these separate softmaxes* (Eq. (13)). Intuitively, an image is flagged OOD when it attains non-negligible confidence at the coarse level but fails to concentrate probability on any subclass of that parent. The method is purely zero-shot and relies only on the generated semantic hierarchy and CLIP's pre-trained embeddings. Below, we detail each component of the pipeline.

### 3.1 HIERARCHICAL PROMPT-AWARE SEMANTIC LABELS

To enrich the semantic structure of the ID label space $\mathcal{Y}^C = \{y_i^C\}_{i=1}^N$, we construct sets of Hierarchical Prompt-Aware Semantic Labels (HPSL), which are fine-grained label sets that maps each coarse class $y_i^C$ to a set of semantically coherent subclasses. Inspired by CHiLS Novack et al. (2023), our pipeline generates and filters candidate subclasses using a combination of prompt-based LLMs, lexical vocabularies, and semantic similarity scoring via CLIP.

For each class $y_i^C$, we begin by querying a language model (e.g., GPT-4) with a structured prompt[1] of the form:

```
"Generate a list of m subtypes that belong to the category
                  [context]: class-name"
```

This produces a candidate set $\mathcal{Y}_{y_i^C}^g$ of up to $m$ textual labels to approximate the hierarchical map (see Eq. (6)). The context tag (e.g., animal, vehicle, food) is tailored to the dataset domain, improving relevance and specificity. Further details are provided in Appendix A.1.

However, relying exclusively on LLM outputs may introduce noisy or irrelevant suggestions. Figure 2 illustrates this process, where subclass candidates from the LLM are supplemented and filtered using a global lexical vocabulary $\mathcal{Y}^v$ for all superclasses $y_i^C$, which consists of curated nouns extracted from external resources such as WordNet Miller (1994)[2]. We use the same fixed vocabulary $\mathcal{Y}^v$ for *all* superclasses $y_i^C$. These sources provide high-quality, domain-aligned labels that serve two purposes:

---

[1]The prompt format can be adjusted to align with the target context or domain.

[2]Alternative lexical resources can also be used.

(*i*) recovering valid hyponyms that the LLM might omit and (*ii*) pruning noisy LLM-generated entries with these additional nouns. The combined candidate pool is then formed as the union:

$$\mathcal{Y}^p_{y^C_i} = \mathcal{Y}^g_{y^C_i} \cup \mathcal{Y}^v. \tag{3}$$

To assess the semantic proximity of each candidate $y^p_j \in \mathcal{Y}^p_{y^C_i}$ to the ID label space, we encode its textual prompt template using CLIP's text encoder $\mathbf{f}^{\text{text}}$, yielding a normalized embedding $\boldsymbol{e}^p_j \in \mathbb{R}^D$. The semantic distance $d_j$ between a candidate subclass and the ID label set is computed as the $\eta$-th percentile of the set of cosine distances between its embedding and all ID superclass embeddings:

$$d_j = \text{percentile}_\eta \left( \{ 1 - \cos(\boldsymbol{e}^p_j, e^C_k) \}^N_{k=1} \right). \tag{4}$$

We retain the top $M$ candidates with the smallest $d_j$ values as the final set of subclasses:

$$\mathcal{Y}^S_{y^C_i} = \text{topk} \Big( \{ d_j \},\ \mathcal{Y}^p_{y^C_i},\ M \Big), \quad \text{where } M \le m + |\mathcal{Y}^v|. \tag{5}$$

This selection process ensures that each subclass is semantically close to the original class while remaining diverse enough to provide fine-grained coverage.

The result is a deterministic mapping function

$$H : y^C_i \mapsto \mathcal{Y}^S_{y^C_i}, \tag{6}$$

We define $H^{-1}$ as the inverse mapping function that retrieves the superclass of a given subclass label.

## 3.2 COARSE-TO-FINE AGREEMENT SCORE

Given an input image $\mathbf{x}$, we first compute its visual representation using CLIP's image encoder, yielding a normalized image embedding $\boldsymbol{h} \in \mathbb{R}^D$ Radford et al. (2021). For each label $y$ in the hierarchy, we precompute its corresponding text embedding using CLIP's text encoder and a fixed prompt template. We denote by $\boldsymbol{e}_i = \mathbf{f}^{\text{text}}(\text{prompt}(y^C_i)) \in \mathbb{R}^D$ the normalized text embedding of superclass $y^C_i$, and by $\tilde{\boldsymbol{e}}_j = \mathbf{f}^{\text{text}}(\text{prompt}(y^S_j)) \in \mathbb{R}^D$ the embedding of subclass $y^S_j$.

Let $\mathcal{Y}^C = \{ y^C_i \}^N_{i=1}$ be the set of superclass (coarse-level) labels, and let $H(y^C_i) = \mathcal{Y}^S_{y^C_i}$ denote the set of fine-grained subclass labels associated with superclass $y^C_i$.

For each label $y$, we define its similarity score with the image embedding $\boldsymbol{h}$ as:

$$\ell(y) = \cos(\boldsymbol{h}, \mathbf{f}^{\text{text}}(\text{prompt}(y))), \tag{7}$$

where $y$ corresponds either to a superclass $y^C_i$ or a subclass $y^S_j$.

At the superclass level, we compute the probability distribution via softmax over all coarse labels:

$$P^{\text{sup}}_i = \frac{\exp\left(\cos(\boldsymbol{h}, \boldsymbol{e}_i)/\tau\right)}{\sum^N_{k=1} \exp\left(\cos(\boldsymbol{h}, \boldsymbol{e}_k)/\tau\right)}. \tag{8}$$

where $\tau > 0$ is a temperature scaling parameter. We define the top superclass confidence as:

$$s_{\text{sup}} = \max_i P^{\text{sup}}_i. \tag{9}$$

At the fine-grained level, we compute similarity scores *within each parent* $y^C_i$ as a parent-conditional softmax:

$$P^{\text{sub}}_{j|i} = \frac{\exp\left(\cos(h, \tilde{\boldsymbol{e}}_j)/\tau\right)}{\sum_{k \in H(y^C_i)} \exp\left(\cos(h, \tilde{\boldsymbol{e}}_k)/\tau\right)}. \tag{10}$$

**Reweighting Subclass Probabilities.** To encourage semantic consistency, each subclass probability is reweighted by the probability of its parent superclass. For a subclass $y^S_j$ whose parent is $y^C_i = H^{-1}(j)$, the final subclass score becomes:

$$\tilde{P}_j = P^{\text{sup}}_i \cdot P^{\text{sub}}_{j|i}. \tag{11}$$

By construction, $\sum_i \sum_{j \in H(y_i)} \tilde{P}_j = \sum_i P_i^{\mathrm{sup}} = 1$.

This follows the CHiLS strategy Novack et al. (2023), where subclass scores are modulated by the confidence in their associated coarse class. Motivated by recent findings Minderer et al. (2021); Kadavath et al. (2022) showing that LLMs, including CLIP's text encoder, tend to be well-calibrated and assign higher confidence to correct outputs. This penalizes mismatched hierarchical assignments and promotes alignment between coarse and fine predictions. The fine-level confidence is given by:

$$s_{\mathrm{sub}} = \max_{j \in \mathcal{Y}^S} \tilde{P}_j, \qquad \text{with } \mathcal{Y}^S = \bigcup_{i=1}^{N} H(y_i^C). \tag{12}$$

**HiPOOD Score.** We define the OOD score as a function of the normalized ratio between the best reweighted fine score and the best coarse score:

$$S_{\mathrm{HiPOOD}}(\mathbf{x}) = \frac{s_{\mathrm{sub}}}{s_{\mathrm{sup}} + s_{\mathrm{sub}}}. \tag{13}$$

Lower values indicate coarse–fine disagreement (typical of OOD samples), whereas values closer to 0.5 suggest hierarchical agreement (typical of ID inputs).

### 3.3 Theoretical Justification of the HiPOOD Score

The score $S_{\mathrm{HiPOOD}}(\mathbf{x})$ measures hierarchical consistency by contrasting the best coarse-level confidence $s_{\mathrm{sup}}$ with the best reweighted fine-level confidence $s_{\mathrm{sub}}$ (see Sec. 3.2). By construction of the parent-conditional fine softmax and the subsequent reweighting, we have $0 < s_{\mathrm{sub}} \leq s_{\mathrm{sup}} \leq 1$. From Eq. (13) it follows that

$$0 < S_{\mathrm{HiPOOD}}(\mathbf{x}) \leq \tfrac{1}{2}, \qquad S_{\mathrm{HiPOOD}}(\mathbf{x}) = \tfrac{1}{2} \iff s_{\mathrm{sub}} = s_{\mathrm{sup}}.$$

Equivalently, defining $\Delta = \log s_{\mathrm{sup}} - \log s_{\mathrm{sub}} \geq 0$,

$$S_{\mathrm{HiPOOD}}(\mathbf{x}) = \frac{s_{\mathrm{sub}}}{s_{\mathrm{sup}} + s_{\mathrm{sub}}} = \frac{1}{1 + \exp(\Delta)} = \sigma(-\Delta), \tag{14}$$

where $\sigma(\cdot)$ denotes the logistic sigmoid. This quantity is a monotonically decreasing function of $\Delta$. Thus, $S_{\mathrm{HiPOOD}}$ is large (near 0.5) when the coarse and fine predictions agree ($\Delta$ close to 0), and small when they disagree (large $\Delta$).

**ID Consistency.** For in-distribution inputs, the most confident parent typically aligns with at least one of its subclasses, so $s_{\mathrm{sub}} \approx s_{\mathrm{sup}}$ and $S_{\mathrm{HiPOOD}}(\mathbf{x}) \approx 0.5$.

**Fine-Grained OOD.** For a novel subclass under a known parent, the coarse score $s_{\mathrm{sup}}$ may remain high, while the mass at the fine level spreads across incorrect subclasses, yielding $s_{\mathrm{sub}} \ll s_{\mathrm{sup}}$ and therefore $S_{\mathrm{HiPOOD}}(\mathbf{x}) \ll 0.5$.

**Coarse-Level OOD.** When no known parent fits, both $s_{\mathrm{sup}}$ and $s_{\mathrm{sub}}$ are low, but the reweighted fine maximum remains dominated by the coarse one ($s_{\mathrm{sub}} \leq s_{\mathrm{sup}}$), again producing $S_{\mathrm{HiPOOD}}(\mathbf{x})$ close to 0.

Linderman et al. (2023) show that coarse-to-fine path-based modeling improves robustness. $S_{\mathrm{HiPOOD}}$ leverages this structure inversely—as a diagnostic for distributional shift. Its formulation encodes the intuitive behavior observed in hierarchical cognition and models: when uncertain, humans and models tend to fall back to broader categories rather than making unreliable fine-grained predictions (cf. Rosch et al. (1976)). Thus, $S_{\mathrm{HiPOOD}}(\mathbf{x})$ offers a mathematically grounded and interpretable indicator of OOD uncertainty: values near 0.5 denote coarse–fine agreement typical of ID, whereas small values approaching 0 flag OOD due to hierarchical disagreement.

## 4 Experiments

### 4.1 Setup

**Datasets and Evaluation Metrics.** We evaluate our method using ImageNet-1K Huang et al. (2021) as the ID dataset. For OOD detection, we use four established benchmarks: iNaturalist Van Horn

Table 1: Performance (%) of HiPOOD variant *in-d5* and competing OOD detection methods. All approaches use the CLIP ViT-B/16 model, with ImageNet-1K as the ID dataset. Arrows indicate the preferred direction for each metric: ↑ denotes higher is better, while ↓ indicates lower is better.

| OOD Dataset | iNaturalist | | SUN | | Places | | Textures | | Average | |
|---|---|---|---|---|---|---|---|---|---|---|
| Metric | AUROC↑ | FPR95↓ | AUROC↑ | FPR95↓ | AUROC↑ | FPR95↓ | AUROC↑ | FPR95↓ | AUROC↑ | FPR95↓ |
| **Requires training (or w. fine-tuning)** | | | | | | | | | | |
| MSP | 87.44 | 58.36 | 79.73 | 73.72 | 79.67 | 74.41 | 79.69 | 71.93 | 81.63 | 69.61 |
| ODIN | 94.65 | 30.22 | 87.17 | 54.04 | 85.54 | 55.06 | 87.85 | 51.67 | 88.80 | 47.75 |
| Energy | 95.33 | 26.12 | 92.66 | 35.97 | 91.41 | 39.87 | 86.76 | 57.61 | 91.54 | 39.89 |
| GradNorm | 72.56 | 81.50 | 72.86 | 82.00 | 73.70 | 80.41 | 70.26 | 79.36 | 72.35 | 80.82 |
| ViM | 93.16 | 32.19 | 87.19 | 54.01 | 83.75 | 60.67 | 87.18 | 53.94 | 87.82 | 50.20 |
| KNN | 94.52 | 29.17 | 92.67 | 35.62 | 91.02 | 39.61 | 85.67 | 64.35 | 90.97 | 42.19 |
| VOS | 94.62 | 28.99 | 92.57 | 36.88 | 91.23 | 38.39 | 86.33 | 61.02 | 91.19 | 41.32 |
| NPOS | 96.19 | 16.58 | 90.44 | 43.77 | 89.44 | 45.27 | 88.80 | 46.12 | 91.22 | 37.93 |
| CATEX | 97.88 | 10.18 | 92.83 | 33.87 | 90.48 | 41.43 | 92.73 | 33.17 | 93.48 | 29.66 |
| **Zero-shot (no training required)** | | | | | | | | | | |
| Mahalanobis | 55.89 | 99.33 | 59.94 | 99.41 | 65.96 | 98.54 | 64.23 | 98.46 | 61.51 | 98.94 |
| Energy | 85.09 | 81.08 | 84.24 | 79.02 | 83.38 | 75.08 | 65.56 | 93.65 | 79.57 | 82.21 |
| ZOC | 86.09 | 87.30 | 81.20 | 81.51 | 83.39 | 73.06 | 76.46 | 98.90 | 81.79 | 85.19 |
| MCM | 94.59 | 32.20 | 92.25 | 38.80 | 90.31 | 46.20 | 86.12 | 58.50 | 90.82 | 43.96 |
| CLIPN | 95.27 | 23.94 | 93.93 | 26.17 | **92.28** | **33.45** | **90.93** | **40.82** | 93.10 | 31.10 |
| NegLabel | **99.49** | **1.91** | 95.49 | 20.53 | 91.64 | 35.59 | 90.22 | 43.56 | 94.21 | 25.40 |
| Ours (in-d5) $M = 10$ | 99.40 | 9.26 | **95.78** | **20.49** | 92.19 | 32.69 | 90.05 | 40.83 | **94.31** | 25.82 |

et al. (2018), SUN Xiao et al. (2010), Places Zhou et al. (2018), and Textures Cimpoi et al. (2014). These datasets cover a broad range of natural, indoor, synthetic, and scene-based images, offering complementary visual distributions for comprehensive OOD evaluation. Importantly, we ensure no label overlap between ID and OOD categories in our hierarchies. ImageNet-1K provides a natural fit for hierarchical modeling, as its class ontology is derived from WordNet Miller (1994), which organizes concepts into a structured semantic tree. This inherent hierarchy aligns well with our proposed framework, which exploits superclass–subclass relationships for zero-shot OOD detection.

Our evaluation focuses on two widely adopted metrics in the OOD detection literature: the Area Under the Receiver Operating Characteristic curve (AUROC) and the False Positive Rate at 95% True Positive Rate (FPR95). The AUROC measures the overall separability between ID and OOD samples across varying thresholds, with higher values indicating better performance. FPR95 reflects the rate at which OOD inputs are incorrectly accepted when 95% of ID samples are correctly classified, with lower values indicating stronger robustness to OOD inputs.

**Implementation Details.** Our zero-shot OOD detection framework is built upon the CLIP architecture Radford et al. (2021), using the widely adopted ViT-B/16 variant due to its balance between performance and efficiency. For subclass label expansion, we leverage WordNet Miller (1994) as the lexical source. The number of subclass candidates per superclass is tuned according to the cardinality of the coarse label set to ensure a sufficiently broad semantic search space. In practice, we set $m$ and $M$ within the range $[10, 500]$ (see Appendix B.9), favoring higher values when the number of ID classes is small to increase coverage and semantic richness. In our main experiments we use GPT-4 to propose subclasses, but Appendix B.8 shows that replacing it with open-source LLMs (Llama-3.1, Mistral-Large, Qwen2) only slightly reduces performance, indicating that HiPOOD is robust to the choice of LLM. We also test an alternative Wikipedia-derived vocabulary; this slightly decreases performance but preserves the overall trends, confirming that curated lexical resources are helpful but not strictly required (see Appendix A.2).

We apply semantic filtering based on CLIP text embeddings by computing the $\eta$-th percentile of cosine distances with $\eta = 0.05$, and we use a softmax temperature of $\tau = 0.01$ during scoring to sharpen distributional confidence (see Appendix B.10). The NPOS protocol (Tao et al., 2023) was adopted for OOD methods on fine-tuned CLIP, where the last two blocks of CLIP's image encoder are fine-tuned, and a fully connected layer is appended to map image features to class predictions. All experiments are executed on a single NVIDIA GeForce RTX 4090 GPU.

## 4.2 MAIN RESULTS

We compare the performance of HiPOOD against several established OOD detection approaches in Table 1. Despite operating in a zero-shot setting, HiPOOD consistently outperforms both classical post-hoc methods (MSP Hendrycks & Gimpel (2018), ODIN Liang et al. (2018), GradNorm Huang et al. (2021), KNN Sun et al. (2022), VOS Du et al. (2022)), NPOS Tao et al. (2023), CATEX Liu et al. (2023) and zero-shot baselines (Mahalanobis Lee et al. (2018), Energy Liu et al. (2020), ZOC Esmaeilpour et al. (2022), MCM Ming et al. (2022), CLIPN Wang et al. (2023), NegLabel Jiang

et al. (2024)). In particular, it achieves results on par with or superior to NegLabel, while requiring only a fraction of the additional label space. These outcomes highlight the effectiveness of our hierarchical prompt-aware formulation.

**ImageNet Hierarchy Depth.** We evaluate five HiPOOD variants, denoted *in-d1* to *in-d5* corresponding to different depths in the ImageNet hierarchy, as defined in BREEDS Santurkar et al. (2020). These settings span from coarse-grained ID groupings (e.g., 2 superclasses in *in-d1*) to fine-grained leaf-level classes (460 superclasses in *in-d5*). Intermediate settings contain 10 (*in-d2*), 30 (*in-d3*), and 128 (*in-d4*) coarse classes, allowing us to assess how label granularity impacts OOD detection (see Table 2). Across all depths, HiPOOD maintains strong AUROC performance, ranging from 95.46% in *in-d1* to 94.31% in *in-d5*, with only a minor drop despite increasing granularity. This stability demonstrates the robustness of our hierarchical score. At the same time, finer hierarchy levels yield lower FPR95 values: from 36.38% in *in-d1* down to 25.82% in *in-d5*, indicating enhanced precision in rejecting OOD samples. This trend confirms that hierarchical supervision benefits OOD detection both at the coarse and fine levels using a VLM. Coarse superclasses provide broad generalization and high recall (high AUROC), while fine-grained subclasses refine predictions and reduce false positives (low FPR95). The HiPOOD score effectively balances both: it flags inputs with inconsistent superclass-subclass confidence, which frequently signals OOD behavior.

Table 2: Performance (%) of HiPOOD (ours) variants, *in-d1* to *in-d4* correspond to increasing levels of hierarchy depth, with 2, 10, 30, and 128 superclasses respectively. The number of subclass candidates per superclass $M$ is tuned based on the depth to ensure broad but controlled coverage, ranging from $M = 500$ (in-d1) to $M = 15$ (in-d4).

| OOD Dataset | iNaturalist | | SUN | | Places | | Textures | | Average | |
|---|---|---|---|---|---|---|---|---|---|---|
| Metric | AUROC↑ | FPR95↓ | AUROC↑ | FPR95↓ | AUROC↑ | FPR95↓ | AUROC↑ | FPR95↓ | AUROC↑ | FPR95↓ |
| Ours (in-d1) $M = 500$ | 99.63 | 26.36 | 96.38 | 33.26 | 93.40 | 38.54 | 92.44 | 47.36 | 95.46 | 36.38 |
| Ours (in-d2) $M = 100$ | 99.51 | 25.12 | 95.65 | 31.69 | 91.94 | 37.38 | 91.76 | 45.03 | 94.72 | 34.81 |
| Ours (in-d3) $M = 50$ | 99.14 | 20.67 | 95.13 | 28.66 | 92.23 | 34.48 | 90.76 | 43.97 | 94.32 | 29.95 |
| Ours (in-d4) $M = 15$ | 99.37 | 16.10 | 94.85 | 26.13 | 91.79 | 35.76 | 90.10 | 41.35 | 94.03 | 29.84 |

Table 3: Zero-shot OOD detection performance (Average AUROC↑ / FPR95↓) across ID datasets with different variants of HiPOOD.

Table 4: Backbone comparison on ImageNet-1K (ID): Average AUROC↑ / FPR95↓. Rows are hierarchy depths and columns are CLIP backbones.

| Depth | HiLLM | HiONTO | HiPOOD w/o RW |
|---|---|---|---|
| in-d1 | 93.04 / 43.02 | 93.53 / 41.59 | 90.58 / 46.44 |
| in-d2 | 92.29 / 41.45 | 92.78 / 40.02 | 89.84 / 44.86 |
| in-d3 | 91.89 / 38.59 | 92.38 / 37.16 | 88.71 / 43.52 |
| in-d4 | 91.61 / 36.48 | 92.09 / 35.05 | 88.42 / 41.41 |
| in-d5 | 91.93 / 32.46 | 92.42 / 31.03 | 88.75 / 37.40 |

| Depth | ResNet50 | ViT-B/16 | ViT-B/32 | ViT-L/14 |
|---|---|---|---|---|
| in-d1 | 94.82 / 38.80 | 95.46 / 36.38 | 95.17 / 38.60 | 96.54 / 31.40 |
| in-d2 | 94.39 / 34.15 | 94.72 / 34.81 | 94.50 / 34.65 | 94.76 / 35.30 |
| in-d3 | 94.49 / 29.46 | 94.32 / 29.95 | 94.75 / 26.32 | 94.35 / 32.25 |
| in-d4 | 92.51 / 38.41 | 94.03 / 29.84 | 92.23 / 39.20 | 93.14 / 36.14 |
| in-d5 | 93.25 / 28.45 | 94.21 / 25.82 | 93.29 / 29.69 | 94.16 / 27.18 |

## 4.3 ABLATION STUDY

We conduct an ablation study on the main components of our framework (cf. Appendix B.6).

**Ablation across hierarchy depth.** Table 3 reports mean zero-shot OOD performance with variability for three HiPOOD variants across ImageNet hierarchy depths (in-d1–in-d5). Here, *HiLLM* denotes a hierarchy generated directly by a large language model without vocabulary filtering, *HiONTO* reuses the predefined ImageNet hierarchy, and *HiPOOD w/o RW* removes the parent reweighting step. Two trends are clear. First, *HiONTO* consistently outperforms *HiLLM* at every depth, with gaps that exceed the run-to-run standard deviation, indicating that taxonomy-grounded labels reduce noise and lower false positives. Second, *HiPOOD w/o RW* is consistently weaker, with degradations well beyond the reported variability, confirming the role of parent conditioning in maintaining coarse–fine coherence. As depth increases (toward in-d5), FPR95 decreases for all variants while AUROC remains comparatively stable within one standard deviation, suggesting that finer subclass sets primarily improve specificity without harming discrimination.

**Backbone Sensitivity.** We evaluate HiPOOD with CLIP backbones and report mean AUROC/FPR95 averaged over ImageNet hierarchy depths in Table 4. HiPOOD maintains high AUROC across architectures: ViT-L/14 attains the best AUROC at coarse depth (in-d1, up to 96.54%), while ViT-B/16 yields the lowest average FPR95 at fine depth (in-d5, 25.82%). The method generalizes well to both ResNet and ViT variants; even ResNet50 remains competitive (e.g., 94.82% AUROC at

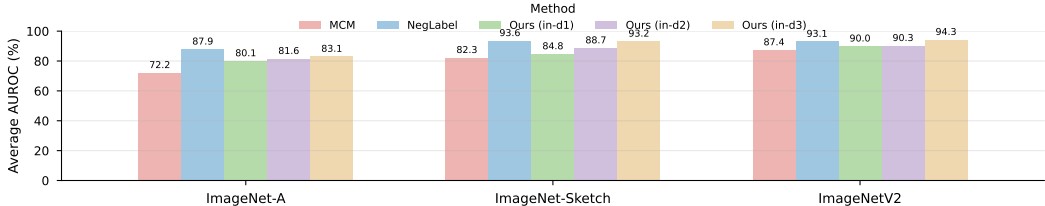

Figure 3: Average AUROC (%) across domain-shifted ImageNet-* variants for different zero-shot OOD detection methods. A higher AUROC indicates better performance.

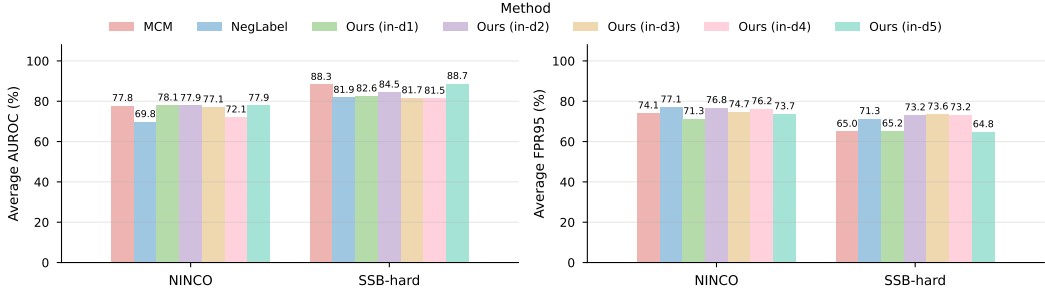

Figure 4: Average AUROC (%) on hard OOD datasets for different zero-shot OOD detection methods. A higher AUROC indicates better performance.

in-d1). This supports prior observations (e.g., MCM Ming et al. (2022)) that strong vision–language alignment, rather than architecture-specific tuning, largely drives performance.

### 4.4 ROBUSTNESS TO DOMAIN SHIFT AND HARD OOD

Figure 3 summarizes average AUROC across three domain-shifted ImageNet variants (Sketch, A, V2) and shows three consistent trends. First, the hierarchical depth matters: our method improves monotonically from in-d1 to in-d3 on all domains. Second, the hierarchical approach closes or exceeds strong flat baselines: on Sketch, in-d3 nearly matches NegLabel, and on V2 it surpasses it; on ImageNet-A, a gap remains to NegLabel but the gains over MCM are substantial. These results indicate that enriching labels with hierarchical structure yields robust improvements under distribution shift, with larger benefits at greater depth. More can be found in Appendix B.1.

Beyond these domain shifts, we also evaluate HiPOOD on two hard OOD benchmarks, NINCO Bitterwolf et al. (2023) and SSB-hard Vaze et al. (2022) (see Figure 4). On NINCO, the best configuration of HiPOOD (in-d1) slightly improves both AUROC and FPR95 over MCM, while clearly outperforming NegLabel. On SSB-hard, HiPOOD (in-d5) achieves the highest AUROC and lowest FPR95 among all methods, again surpassing both MCM and NegLabel. These results indicate that hierarchical consistency remains effective even in challenging near-OOD regimes where standard flat zero-shot baselines struggle.

### 4.5 DISCUSSION

The standard flat-label zero-shot baseline (MCM) scores images only against the ID label list, without hierarchical structure. More recent methods such as CLIPN and NegLabel augment this flat setting with handcrafted negative prompts or surrogate labels to approximate the OOD space. While effective, these negatives can suffer from incomplete coverage and semantic drift, especially on ambiguous or novel inputs. HiPOOD takes a different path: it does not attempt to model the OOD distribution directly, but instead exploits *subsumption* between superclasses and subclasses and flags samples that break coarse–fine semantic consistency within the known ID hierarchy. In terms of label budget, NegLabel may require up to 10,000 external labels to be competitive. HiPOOD uses a much smaller, ID-grounded expansion with no more than 500 fine-grained subclass candidates derived from the ID labels themselves. This yields a lightweight and semantically principled label set that remains constrained to observed categories while improving robustness over the flat MCM baseline. Concretely, under comparable total label budgets HiPOOD improves over NegLabel by up to +3.9 AUROC and about −6 FPR95 (App. B.5), and it still exceeds the supervised hierarchical CATEX baseline by roughly +0.8 AUROC / −3.8 FPR95 on average despite using no ID images (App. B.4). These results show that HiPOOD offers a uniquely efficient and principled alternative: a zero-shot OOD detector that leverages hierarchical consistency to outperform flat baselines and rival supervised hierarchical methods while using substantially fewer external labels.

## 5 RELATED WORK

**Post-hoc OOD Methods.**  Early work exposed the overconfidence of deep networks on unknowns (Hendrycks & Gimpel, 2018), prompting scores computed on a fixed model to separate in- from out-of-distribution inputs. Confidence- and logit-based detectors include MSP/ODIN (Hendrycks & Gimpel, 2018; Liang et al., 2018), Mahalanobis distance in feature space (Lee et al., 2018), energy-based scores (Liu et al., 2020), gradient norms (Huang et al., 2021), activation rectification (ReAct) (Sun et al., 2021), deep $k$NN (Sun et al., 2022), sparse activation shaping (ASH) (Djurisic et al., 2023), and ViM (Wang et al., 2021). These methods are simple and strong under closed-set assumptions, but can degrade when unknowns are semantically close to ID or when label spaces are large and structured.

**Training-time regularization and outlier exposure.**  A complementary line shapes decision boundaries during training via confidence penalties, energy shaping, or auxiliary data. Outlier Exposure encourages low confidence on external data (Hendrycks et al., 2018); virtual outlier synthesis (VOS) generates pseudo-unknowns to regularize the margin (Du et al., 2022); other works leverage Bayesian/ensembles for epistemic uncertainty (Gal & Ghahramani, 2016; Lakshminarayanan et al., 2017; Maddox et al., 2019; Malinin & Gales, 2018; Wen et al., 2020) or stabilize logits (e.g., LogitNorm) (Wei et al., 2022). While effective, these approaches require retraining and are sensitive to the coverage of the OOD proxy.

**Zero-shot OOD with vision–language models.**  Pre-trained vision–language models (VLMs) such as CLIP (Radford et al., 2021) enable zero-shot detection by comparing images to natural-language labels. Representative flat-label approaches include MCM (Ming et al., 2022) and ZOC (Esmaeilpour et al., 2022). Negative prompting/label expansion improves rejection: CLIPN (Wang et al., 2023) and NegLabel (Jiang et al., 2024) teach CLIP to "say no" using curated negatives. Recent variants refine negative construction or class centers (e.g., CSP (Chen et al., 2024), TagOOD (Li et al., 2024), AdaNeg (Zhang & Zhang, 2024)) and explore LLM-guided outlier exposure (Cao et al., 2024). These methods remain largely *flat*.

**Hierarchical OOD methods.**  A few works explicitly couple class hierarchies and OOD detection. ProHOC Wallin et al. (2025) and the hierarchical framework of Linderman et al. (2023) train multi-depth visual networks or node-specific heads over a given tree, with loss terms and thresholds tuned on ID data to decide where to stop in the hierarchy. CATEX (Liu et al., 2023) learns hierarchical context descriptions that act as *learnable prompts* attached to each category, improving OOD performance but requiring supervised optimization and calibration. In contrast, HiPOOD uses only a pre-trained VLM with fixed, human-readable prompts and converts coarse–fine agreement over a text-induced hierarchy into a single training-free OOD score.

**Large-scale evaluation and domain shift.**  Modern evaluations stress large label spaces and realistic shifts (e.g., ImageNet-Sketch, ImageNet-A, ImageNetV2) with threshold-free metrics such as AUROC and FPR@95. VLM-based zero-shot methods scale naturally but can be brittle on near-OOD. By leveraging a hierarchy over ID labels and scoring coarse–fine consistency, HiPOOD improves separation in these challenging regimes while remaining training-free and label-efficient.

## 6 CONCLUSION

We introduced HiPOOD, a zero-shot OOD detection framework that uses hierarchical, prompt-based representations over known ID labels. Our method captures consistency violations between super- and subclass predictions, yielding an interpretable, training-free scoring function. Through prompt-driven subclass generation and lexical filtering with CLIP, we construct label sets without auxiliary negatives. HiPOOD delivers competitive zero-shot performance, including a comparison to MCM under flat labels and additional results against NegLabel. Unlike NegLabel, which relies on a large number of auxiliary labels to converge, HiPOOD leverages intrinsic information about the ID space via the subsumption relationship. Because many fields exhibit rich hierarchical structure, our approach adapts readily across domains while reinforcing open-world robustness for open-vocabulary models. Overall, these contributions establish HiPOOD as an efficient alternative to both flat and supervised hierarchical approaches for zero-shot OOD detection.

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

## A  SEMANTIC LABEL HIERARCHIES GENERATION

### A.1  PROMPT GENERATION TEMPLATES

To generate semantically meaningful subclass candidates for each in-distribution (ID) label, we used structured prompts formulated for large language models (LLMs). These prompts were designed to elicit fine-grained types or subcategories that belong to a given high-level concept, ensuring consistency with the visual and semantic domains of our ID classes.

A key design choice was to explicitly control the number of subclass candidates requested from the LLM. For each parent label, we specified a target number $m$ of subtypes to be generated. This ensured a uniform and controllable subclass size across the hierarchy, facilitating both computational tractability and semantic coverage.

The final prompt template we adopted was:

```
Generate a list of m subtypes that belong to the
category [context]:  <label>
```

Here:

- `<label>` is the ID class name (e.g., `dog`, `vehicle`, `fruit`),
- `[context]` is a dataset-specific token (e.g., food, animal, artifact), based on the CHiLS method Novack et al. (2023),
- $m$ is the number of desired subclasses (e.g., $m = 10$).

We experimented with several variations. Examples include:

- `List m types of <label>.`
- `Name m different categories under the concept of <label>.`
- `Give a list of m subclasses or fine-grained examples of <label>.`

All prompts were submitted to GPT-4, with a temperature of 0.7 and a maximum token limit adapted to allow for up to $m = 10$ valid candidates per label. The full list of candidates was later refined using lexical and embedding-based filtering (see Sections A.2 and A.3).

**Example Generation.**  As an illustration, for the ID class `car` from the vehicle category, we submitted the following prompt:

```
Generate a list of 10 subtypes that belong to the
category [vehicle]:  car.  Return a comma-separated
list.
```

The LLM (GPT-4) responded with:

```
sedan, coupe, convertible, hatchback, SUV, station
wagon, pickup truck, sports car, minivan, roadster
```

This list satisfies the constraints we imposed:

- It is a flat, comma-separated list of noun phrases;
- All terms refer to concrete, visually distinguishable subclasses of the parent concept `car`;
- The subclass embeddings are semantically close to the parent class in the CLIP embedding space (validated in A.3).

### A.2  VOCABULARY FILTERING

Although GPT-4 is effective in generating plausible subcategories for a given concept, its output may include noisy or linguistically malformed candidates, especially when queried in an unconstrained setting. To ensure that the subclass hierarchy remains semantically coherent and visually usable, we introduce a filtering step based on lexical validity and semantic alignment.

**Motivation.** In early experiments, we observed that many GPT-generated subclass candidates $\mathcal{Y}^g$ were overly descriptive or attribute-based rather than naming distinct visual categories. For example, when querying subclasses of the class `car`, the model might return:

```
fast car, luxury, SUV, electric, big truck, vehicle, sedan,
convertible
```

Here, several terms (*fast car*, *luxury*, *electric*) describe properties or attributes rather than specific object categories. Others (*vehicle*) are hypernyms or too abstract to serve as visual labels. These labels are unsuitable for downstream use in a vision-language model, as they do not correspond to well-defined visual prototypes.

To address this, we use a controlled vocabulary $\mathcal{Y}^v$ of 100,000 curated noun labels extracted from WordNet Miller (1994), designed to capture visually nameable categories. This external vocabulary serves a dual purpose: (i) complementing the LLM outputs by introducing additional valid terms that may be omitted by the model, and (ii) filtering out ambiguous or non-visual labels.

The combined pool of candidate subclasses is obtained via union:

$$\mathcal{Y}^p_{y^C_i} = \mathcal{Y}^g_{y^C_i} \cup \mathcal{Y}^v \tag{15}$$

Each candidate $y^p_j \in \mathcal{Y}^p_{y^C_i}$ is encoded via CLIP's text encoder with the prompt "A photo of a $y^p_j$", producing a normalized embedding $\boldsymbol{e}^p_j \in \mathbb{R}^D$ (all vectors are L2-normalized.). We assess the alignment of this candidate with the overall ID class space by computing a robust semantic distance based on the $\eta$-th percentile of its distances to all ID class embeddings:

$$d_j = \text{percentile}_\eta \left( \left\{ 1 - \cos(\boldsymbol{e}^p_j, \boldsymbol{e}^C_k) \right\}_{k=1}^N \right) \tag{16}$$

This percentile-based metric prioritizes candidates that are consistently semantically close to most ID labels, rather than merely close to one.

We then select the top $M$ candidates with the smallest $d_j$ values to construct the final subclass label set:

$$\mathcal{Y}^S_{y^C_i} = \text{topk} \left( \{d_j\}, \mathcal{Y}^p_{y^C_i}, M \right) \tag{17}$$

This process yields a refined and diverse set of fine-grained labels for each superclass, guaranteeing both lexical validity and semantic relevance. The resulting mapping is formally defined as:

$$H : y^C_i \mapsto \mathcal{Y}^S_{y^C_i} \tag{18}$$

An inverse mapping $H^{-1}(y^S_j)$ also exists and allows recovering the parent class of a given subclass.

The full procedure is summarized in Algorithm 2; its output mapping $H(y^C_i)$ (parent $\rightarrow$ filtered subclasses) is then used at inference by the parent-conditional softmax and reweighting (see Sec. 3.2) to compute $s_{\text{sub}}$ and the HiPOOD score.

**Handling leaf-like classes.** For ID labels that are already very fine-grained or admit no reliable subclasses ("leaf-like" classes), we apply a simple back-off strategy: we add the parent label itself as a pseudo-subclass, and we suffix all retained candidates with the parent name (e.g., "night traffic light", "broken traffic light").

## A.3 PROMPT ENSEMBLING FOR TEXT EMBEDDINGS

To obtain more robust and representative text embeddings for both ID classes and their subclasses, we follow the prompt ensembling strategy introduced by Radford et al. (2021). Instead of relying on a single prompt (e.g., "a photo of a <label>"), we average the text representations obtained from a diverse set of natural language templates.

---

**Algorithm 2:** Hierarchical Prompt-Aware Semantic Labels Construction (Detailed Version)

---

**Input**: ID label set $\mathcal{Y}^C = \{y_i^C\}_{i=1}^N$, CLIP text encoder $\mathbf{f}^{\text{text}}$, LLM-generated subclasses $\mathcal{Y}_i^g$, vocabulary nouns $\mathcal{Y}^v$, number of subclasses $M$

**Output**: Subclass mapping $H : y_i^C \mapsto \mathcal{Y}_{y_i^C}^S$

1: /* Extract text embeddings of ID labels */
2: **for** each $y_i^C \in \mathcal{Y}^C$ **do**
3:     $e_i^C \leftarrow \mathbf{f}^{\text{text}}(\text{prompt}(y_i^C)/\|\mathbf{f}^{\text{text}}(\text{prompt}(y_i^C)\|_2$
4: **end for**
5: **for** each $y_i^C \in \mathcal{Y}^C$ **do**
6:     /* Merge LLM and vocabulary labels */
7:     $\mathcal{Y}_{y_i^C}^p \leftarrow \mathcal{Y}_{y_i^C}^g \cup \mathcal{Y}^v$
8:     /* Encode and filter subclass candidates */
9:     **for** each $y_j^p \in \mathcal{Y}_{y_i^C}^p$ **do**
10:        $e_j^p \leftarrow \mathbf{f}^{\text{text}}(\text{prompt}(y_j^p)/\|\mathbf{f}^{\text{text}}(\text{prompt}(y_j^p))\|_2$
11:        $d_j \leftarrow \text{percentile}_\eta(\{1 - \cos(e_j^p, e_k^C)\}_{k=1}^N)$
12:     **end for**
13:     /* Select $M$ candidates with lowest $d_j$ */
14:     $\mathcal{Y}_{y_i^C}^S \leftarrow \text{topk}(\{d_j\}, \mathcal{Y}_{y_i^C}^p, M)$
15:     $H(y_i^C) \leftarrow \mathcal{Y}_{y_i^C}^S$
16: **end for**
17: **return** $H : y_i^C \mapsto \mathcal{Y}_{y_i^C}^S$

---

We use a fixed list of 80 prompts adapted from the CLIP zero-shot evaluation protocol. Each template includes a placeholder for the class name (e.g., a drawing of a <dog>) and varies in tone, context, and composition to capture richer linguistic and visual associations.

An excerpt of 32 representative prompts selected from the full set of 80 templates is shown in Table 5 and the full list of 80 prompt templates is used during inference. For each class $y \in \mathcal{Y}$, we generate the corresponding texts, encode them using CLIP's text encoder, normalize each embedding, and average the results:

$$e_y^{\text{text}} = \frac{1}{|\mathcal{T}|} \sum_{t \in \mathcal{T}} \frac{\mathbf{f}^{\text{text}}(t(y))}{\|\mathbf{f}^{\text{text}}(t(y))\|_2} \tag{19}$$

where $\mathcal{T}$ is the set of prompt templates, $\mathbf{f}^{\text{text}}$ is the CLIP text encoder, and $t(y)$ denotes the template-conditioned textual input. Here, $y$ corresponds either to a superclass label $y_i^C$ or to a subclass label $y_j^S$, depending on the stage of the HiPOOD pipeline.

**Benefit.** Prompt ensembling reduces sensitivity to prompt phrasing, improves alignment with image embeddings, and leads to smoother and more semantically meaningful embeddings Kim et al. (2024). This is important when measuring subclass-to-class similarity or computing distances in the HiPOOD score. This step is applied uniformly to all labels: ID classes, subclasses, and OOD candidates.

## B ADDITIONAL RESULTS

### B.1 ROBUSTNESS TO DOMAIN SHIFT AND HARD OOD

We assess robustness under three ImageNet-derived domain shifts—Sketch (drawing/style shift), A (natural adversarial images), and V2 (curation shift)—using zero-shot OOD detection with a fixed CLIP-B/16 backbone. Table 6 reports AUROC (higher is better) and FPR@95 (lower is better) on four OOD sets (iNaturalist, SUN, Places, Textures) plus their average. Two patterns emerge. First, hierarchical depth improves performance consistently across domains: moving from *Ours (in-d1)* to *Ours (in-d3)* yields sizable AUROC gains and lower FPR@95 on all three ID variants (e.g., on Sketch the average AUROC rises from 84.77 to 93.24; on ImageNet-A from 80.09 to 83.06; on V2 from 89.97 to 94.35). This supports our hypothesis that finer, parent-conditioned subclass sets

Table 5: Example prompt templates used for text ensembling. Each {} is replaced with a class or subclass label.

```
a photo of a {}.                      a sketch of the {}.
a good photo of the {}.               a painting of a {}.
a photo of many {}.                   a sculpture of a {}.
a photo of one {}.                    a rendering of the {}.
a photo of my {}.                     graffiti of the {}.
the nice {}.                          a tattoo of the {}.
a cropped photo of the {}.            a toy {}.
a blurry photo of the {}.             a cartoon {}.
a black and white photo of a {}.      art of the {}.
a close-up photo of a {}.             a plastic {}.
a bright photo of a {}.               the {} in a video game.
a dark photo of the {}.               a photo of the cool {}.
a low resolution photo of the {}.     a weird photo of a {}.
a photo of the small {}.              a close-up photo of the {}.
a photo of the large {}.              a low resolution photo of a {}.
```

help separate near-OOD by enforcing coarse–fine agreement rather than relying on a flat label space. Second, the hierarchical approach narrows or closes the gap to strong flat baselines: *Ours (in-d3)* nearly matches NegLabel on Sketch and surpasses it on V2 in average AUROC, while remaining clearly stronger than MCM across all domains. On the more challenging ImageNet-A shift, NegLabel still holds an advantage, indicating that curated negative vocabularies can help on hard, natural adversaries; nevertheless, increasing hierarchical depth steadily improves our results without any training or external outlier data. Overall, these findings indicate that hierarchy-aware scoring is a robust, training-free alternative for zero-shot OOD detection under distribution shift, with the largest benefits appearing at greater depth where fine-grained subclass structure better regularizes confidence.

We further test HiPOOD on hard OOD benchmarks, NINCO and SSB-hard, using ImageNet-1K as ID (Table 7). On NINCO, HiPOOD (in-d1, $M{=}500$) attains the best AUROC/FPR@95 (78.06 / 71.29) and improves over both MCM (77.81 / 74.14) and NegLabel (69.82 / 77.09). On SSB-hard, HiPOOD (in-d5, $M{=}10$) achieves the strongest performance, with 88.73 AUROC and 64.82 FPR@95, slightly surpassing MCM (88.32 / 65.05) and clearly outperforming NegLabel (81.87 / 71.32). Averaged over both hard benchmarks, HiPOOD (in-d5) yields the highest AUROC (83.33 vs. 83.07 for MCM), while HiPOOD (in-d1) attains the lowest FPR@95 (68.25 vs. 69.60 for MCM), confirming that hierarchical consistency remains effective even under challenging real-world OOD regimes.

Table 6: Zero-shot OOD detection performance robustness to domain shift. All methods are based on CLIP-B/16. The ID data are ImageNet-* variants. All values are percentages.

| ID Dataset | Method | iNaturalist | | SUN | | Places | | Textures | | Average | |
|---|---|---|---|---|---|---|---|---|---|---|---|
| | | AUROC↑ | FPR95↓ | AUROC↑ | FPR95↓ | AUROC↑ | FPR95↓ | AUROC↑ | FPR95↓ | AUROC↑ | FPR95↓ |
| | MCM | 87.74 | 63.06 | 85.35 | 67.24 | 81.19 | 70.64 | 74.77 | 79.59 | 82.26 | 70.13 |
| | NegLabel | **99.34** | **2.24** | 94.93 | 22.73 | 90.78 | 38.62 | 89.29 | 46.10 | **93.59** | **27.42** |
| ImageNet-Sketch | Ours (in-d1) | 82.44 | 70.46 | 87.68 | 69.88 | 84.11 | 70.92 | 84.84 | 66.19 | 84.77 | 69.36 |
| | Ours (in-d2) | 89.99 | 76.73 | 89.90 | 68.65 | 86.52 | 69.87 | 88.51 | 62.39 | 88.73 | 69.41 |
| | Ours (in-d3) | 91.51 | 57.32 | **95.33** | **33.38** | **92.95** | **39.66** | **93.19** | **36.28** | 93.24 | 41.66 |
| | MCM | 79.50 | 76.85 | 76.19 | 79.78 | 70.95 | 80.51 | 61.98 | 86.37 | 72.16 | 80.88 |
| | NegLabel | **98.80** | **4.09** | **89.83** | **44.38** | **82.88** | **60.10** | **80.25** | **64.34** | **87.94** | **43.23** |
| ImageNet-A | Ours (in-d1) | 86.17 | 65.08 | 82.25 | 58.10 | 78.07 | 66.45 | 73.88 | 67.18 | 80.09 | 64.20 |
| | Ours (in-d2) | 88.06 | 62.73 | 83.70 | 55.38 | 79.29 | 64.51 | 75.37 | 65.62 | 81.61 | 62.06 |
| | Ours (in-d3) | 90.02 | 60.86 | 84.91 | 53.50 | 80.54 | 63.02 | 76.76 | 64.44 | 83.06 | 60.46 |
| | MCM | 91.79 | 45.90 | 89.88 | 50.73 | 86.52 | 56.25 | 81.51 | 69.57 | 87.43 | 55.61 |
| | NegLabel | **99.40** | **2.47** | 94.46 | 25.69 | 90.00 | 42.03 | 88.46 | 48.90 | 93.08 | 29.77 |
| ImageNetV2 | Ours (in-d1) | 88.23 | 61.54 | 92.33 | 50.72 | 89.39 | 54.44 | 89.92 | 51.86 | 89.97 | 54.64 |
| | Ours (in-d2) | 91.35 | 68.29 | 91.42 | 58.67 | 88.27 | 62.15 | 90.13 | 54.45 | 90.29 | 60.89 |
| | Ours (in-d3) | 92.52 | 50.85 | **96.38** | **27.38** | **94.14** | **34.07** | **94.38** | **31.29** | **94.35** | **35.90** |

## B.2  RESULTS ON NON-HIERARCHICAL AND DOMAIN-SPECIFIC ID DATASETS.

We also evaluate HiPOOD on fine-grained, domain-specific datasets that are typically treated as flat label spaces in prior OOD work: CUB-200 (birds) Catherine Wah et al. (2011), Food-101 (recipes) Bossard et al. (2014), and FGVC-Aircraft (aircraft models) Maji et al. (2013). For each ID dataset,

Table 7: Zero-shot OOD detection on hard benchmarks (NINCO, SSB-hard) with ImageNet-1K as ID data. All methods use CLIP-B/16.

| ID Dataset | Method | NINCO | | SSB-hard | | Average | |
|---|---|---|---|---|---|---|---|
| | | AUROC↑ | FPR95↓ | AUROC↑ | FPR95↓ | AUROC↑ | FPR95↓ |
| | MCM | 77.81 | 74.14 | 88.32 | 65.05 | 83.07 | 69.60 |
| | NegLabel | 69.82 | 77.09 | 81.87 | 71.32 | 75.85 | 74.21 |
| | HiPOOD (in-d1, $M{=}500$) | **78.06** | **71.29** | 82.58 | 65.21 | 80.32 | **68.25** |
| ImageNet-1K | HiPOOD (in-d2, $M{=}100$) | 77.89 | 76.80 | 84.50 | 73.17 | 81.20 | 74.99 |
| | HiPOOD (in-d3, $M{=}50$) | 77.13 | 74.72 | 81.66 | 73.61 | 79.40 | 74.17 |
| | HiPOOD (in-d4, $M{=}15$) | 72.14 | 76.23 | 81.47 | 73.23 | 76.81 | 74.73 |
| | HiPOOD (in-d5, $M{=}10$) | 77.92 | 73.72 | **88.73** | **64.82** | **83.33** | 69.27 |

Table 8: Zero-shot inter-dataset OOD detection on fine-grained, domain-specific ID datasets (CLIP-B/16). All values are percentages.

| ID Dataset | Method | iNaturalist | | Places | | SUN | | Textures | | Average | |
|---|---|---|---|---|---|---|---|---|---|---|---|
| | | AUROC↑ | FPR95↓ | AUROC↑ | FPR95↓ | AUROC↑ | FPR95↓ | AUROC↑ | FPR95↓ | AUROC↑ | FPR95↓ |
| | MCM | 98.24 | 9.83 | 99.10 | 4.93 | 98.57 | 6.65 | 98.75 | 6.97 | 98.66 | 7.09 |
| CUB-200 | NegLabel | **99.96** | **0.18** | **99.99** | **0.02** | **99.90** | **0.33** | **99.99** | **0.01** | **99.96** | **0.13** |
| | HiPOOD | 99.77 | 6.16 | 99.59 | 2.10 | 99.62 | 4.30 | 99.92 | 1.23 | 99.73 | 3.45 |
| | MCM | 99.78 | 0.64 | 99.75 | 0.90 | 99.58 | 1.86 | 98.62 | 4.04 | 99.43 | 1.86 |
| Food-101 | NegLabel | **99.99** | **0.01** | **99.99** | **0.01** | **99.99** | **0.01** | 99.60 | 1.61 | **99.90** | **0.40** |
| | HiPOOD | 99.82 | 0.43 | **99.99** | 0.20 | **99.99** | 0.51 | **99.65** | **1.40** | 99.86 | 0.64 |
| | MCM | 99.10 | 3.50 | 99.20 | 2.70 | 98.90 | 3.20 | 98.80 | 3.40 | 99.00 | 3.20 |
| FGVC-Aircraft | NegLabel | **99.97** | **0.10** | **99.99** | 0.80 | **99.93** | **0.20** | 99.95 | **0.15** | **99.96** | **0.12** |
| | HiPOOD | 99.71 | 2.25 | 99.75 | 1.43 | 99.89 | 0.60 | **99.96** | 0.50 | 99.83 | 1.20 |

we use the same four OOD sets (iNaturalist, Places, SUN, Textures) and report average AUROC and FPR95 in Table 8. Across all three domains, HiPOOD consistently improves over the flat zero-shot baseline MCM in both AUROC and FPR95. For example, on CUB-200 the average FPR95 drops from 7.09 (MCM) to 3.45 with HiPOOD, and on FGVC-Aircraft from 3.20 (MCM) to 1.20. Compared to NegLabel, HiPOOD remains competitive despite using a compact, ID-grounded subclass expansion rather than thousands of mined negatives: on Food-101, it reaches 99.86 AUROC / 0.64 FPR95 versus 99.90 / 0.40 for NegLabel, and on CUB-200 it stays within 0.23 AUROC points of NegLabel while substantially improving over MCM. These results suggest that hierarchical prompt-aware reasoning can provide tangible gains even in settings where the label space is not explicitly organized as a taxonomy, while remaining closer to the ID semantics than broad negative-label mining.

### B.3 COMBINING HIERARCHICAL CONSISTENCY WITH NEGATIVE LABELS

We investigate whether the hierarchical HiPOOD score can benefit from explicitly incorporating NegLabel-style negative labels. Throughout this section we keep the notation of Sec. 3: $\mathcal{Y}^C$ denotes the set of coarse ID labels, $H(y_i^C)$ their associated subclasses $\mathcal{Y}_{y_i^C}^S$, and $S_{\text{HiPOOD}}(\mathbf{x})$ the hierarchy-based score defined from $P_i^{\text{sup}}$, $P_{j|i}^{\text{sub}}$ and the reweighted subclass probabilities $\tilde{P}_j$. For NegLabel, we reuse the grouped score $S_{\text{NegLabel}}(\mathbf{x})$ with the same partition of the mined negative labels $\mathcal{Y}^{\text{ng}}$ on the coarse labels per depth into groups $\{G_t\}_{t=1}^{n_g}$ as in the original method, so that all differences stem from the way hierarchical information is injected rather than from implementation details.

**Version 1: hierarchical negative ratio.** The first variant, denoted $S_{\text{HiPOOD-NegLabel}}^{(1)}(\mathbf{x})$, replaces the flat ID mass in NegLabel by the hierarchical mass concentrated on the most likely subclass under each superclass. For each $y_i^C \in \mathcal{Y}^C$ we define

$$s_{\text{sub}}(i) = \max_{y_j^S \in \mathcal{Y}_{y_i^C}^S} \tilde{P}_j, \tag{20}$$

Table 9: Zero-shot OOD detection with the hierarchical negative score (Version 1). All methods use CLIP-B/16.

| ID Dataset | Method | iNaturalist | | SUN | | Places | | Textures | | Average | |
|---|---|---|---|---|---|---|---|---|---|---|---|
| | | AUROC↑ | FPR95↓ | AUROC↑ | FPR95↓ | AUROC↑ | FPR95↓ | AUROC↑ | FPR95↓ | AUROC↑ | FPR95↓ |
| ImageNet-Sketch | HiPOOD+NegLabel (V1, in-d1) | 81.03 | 68.87 | 84.90 | 68.28 | 83.05 | 70.28 | 83.64 | 65.69 | 83.16 | 68.28 |
| | HiPOOD+NegLabel (V1, in-d2) | 88.64 | 75.21 | 87.13 | 66.90 | 85.49 | 69.25 | 87.26 | 61.81 | 87.13 | 68.30 |
| | HiPOOD+NegLabel (V1, in-d3) | 90.16 | 55.80 | 92.56 | 31.63 | 91.92 | 39.04 | 91.94 | 35.70 | 91.64 | 40.55 |
| ImageNet-A | HiPOOD+NegLabel (V1, in-d1) | 84.74 | 63.41 | 79.36 | 56.22 | 76.59 | 65.59 | 72.63 | 66.66 | 78.33 | 62.97 |
| | HiPOOD+NegLabel (V1, in-d2) | 86.58 | 61.06 | 80.78 | 53.51 | 78.06 | 63.74 | 74.13 | 65.06 | 79.89 | 60.84 |
| | HiPOOD+NegLabel (V1, in-d3) | 88.66 | 59.03 | 81.99 | 51.66 | 79.04 | 62.14 | 75.52 | 63.86 | 81.30 | 59.17 |
| ImageNetV2 | HiPOOD+NegLabel (V1, in-d1) | 86.51 | 59.78 | 89.42 | 48.89 | 88.28 | 53.61 | 88.63 | 51.27 | 88.21 | 53.38 |
| | HiPOOD+NegLabel (V1, in-d2) | 89.84 | 66.31 | 88.52 | 57.16 | 86.85 | 61.28 | 88.85 | 53.92 | 88.52 | 59.67 |
| | HiPOOD+NegLabel (V1, in-d3) | 91.14 | 49.05 | 94.14 | 26.17 | 92.97 | 33.18 | 93.14 | 30.68 | 92.85 | 34.77 |

that is, the probability assigned to the best-aligned subclass of $y_i^C$ under the parent-conditioned formulation of HiPOOD. The hierarchical negative score is then

$$S^{(1)}_{\text{HiPOOD-NegLabel}}(\mathbf{x}) = \frac{\displaystyle\sum_{y_i^C \in \mathcal{Y}^C} s_{\text{sub}}(i)}{\displaystyle\sum_{y_i^C \in \mathcal{Y}^C} s_{\text{sub}}(i) + \sum_{t=1}^{n_g} \sum_{y \in G_t} \exp(\ell(y))}, \tag{21}$$

which mirrors the NegLabel ratio but replaces the flat ID term by a hierarchy-aware aggregate. Intuitively, this score rewards images whose probability mass is both concentrated along a consistent parent–child path and clearly separated from the negative labels.

Table 9 reports the resulting performance on the three ImageNet-* domain shifts. On ImageNet-Sketch, increasing depth from in-d1 to in-d3 steadily improves AUROC (from 83.16 to 91.64%) and reduces FPR95 (from 68.28% to 40.55%), closely matching the depth trend of the original HiPOOD score. Similar behaviour is observed on ImageNet-A (78.33/62.97 to 81.30/59.17) and ImageNetV2 (88.21/53.38 to 92.85/34.77). Compared to pure HiPOOD at the same depth, V1 typically trades 1–2 points of AUROC for a modest but consistent gain in FPR95 (around 1 point), indicating that hierarchical negatives make the detector slightly more conservative at high TPR while preserving most of the ranking power.

**Version 2: linear combination.** The second variant, $S^{(2)}_{\text{HiPOOD-NegLabel}}(\mathbf{x})$, keeps the original scores $S_{\text{HiPOOD}}(\mathbf{x})$ and $S_{\text{NegLabel}}(\mathbf{x})$ unchanged and combines them through a simple convex interpolation:

$$S^{(2)}_{\text{HiPOOD-NegLabel}}(\mathbf{x}) = 2\lambda\, S_{\text{HiPOOD}}(\mathbf{x}) + (1-\lambda)\, S_{\text{NegLabel}}(\mathbf{x}), \qquad \lambda \in [0,1]. \tag{22}$$

In all experiments we set $\lambda = 0.5$ to give equal weight to hierarchical agreement and negative-label evidence, without tuning an extra hyperparameter per dataset.

As shown in Table 10, this linear blending behaves, as expected, between the two base detectors. On ImageNet-Sketch, the average AUROC at depth in-d3 reaches 90.48% with FPR95 of 45.64%, which improves over NegLabel alone on this shift but remains below the best hierarchical configuration without negatives. On ImageNet-A and ImageNetV2, V2 again trails the pure HiPOOD score in both AUROC and FPR95 (e.g., 80.53/64.40 vs. 83.06/60.46 on ImageNet-A at in-d3, and 91.48/39.70 vs. 94.35/35.90 on ImageNetV2), suggesting that a naive linear fusion does not consistently dominate either component.

These results indicate that negative labels can provide a small calibration benefit, especially when injected hierarchically as in V1, but the main performance gains of HiPOOD already arise from coarse–fine consistency over ID-grounded subclasses. For simplicity and interpretability, the main paper therefore reports results using $S_{\text{HiPOOD}}(\mathbf{x})$ as the default score, and treats HiPOOD+NegLabel as an optional extension.

### B.4 ALIGNED COARSE ID LABEL SETS AND FAIR COMPARISON

Table 11 reports a fully aligned comparison where all methods share the same coarse ID label sets at depths d1–d5. Across all depths, HiPOOD clearly dominates the flat baselines. At shallow depth d1, HiPOOD reaches an average AUROC of 95.46 with FPR95 = 36.38, compared to 73.35/74.35 for

Table 10: Zero-shot OOD detection with the linear combination score (Version 2, $\lambda = 0.5$). All methods use CLIP-B/16.

| ID Dataset | Method | iNaturalist | | SUN | | Places | | Textures | | Average | |
|---|---|---|---|---|---|---|---|---|---|---|---|
| | | AUROC↑ | FPR95↓ | AUROC↑ | FPR95↓ | AUROC↑ | FPR95↓ | AUROC↑ | FPR95↓ | AUROC↑ | FPR95↓ |
| ImageNet-Sketch | HiPOOD+NegLabel (V2, in-d1) | 78.59 | 75.49 | 84.26 | 72.60 | 81.18 | 74.24 | 83.55 | 69.98 | 81.90 | 73.08 |
| | HiPOOD+NegLabel (V2, in-d2) | 85.61 | 81.67 | 85.84 | 71.69 | 83.90 | 72.87 | 86.25 | 65.90 | 85.40 | 73.03 |
| | HiPOOD+NegLabel (V2, in-d3) | 88.08 | 62.45 | 91.99 | 36.75 | 90.67 | 42.85 | 91.17 | 40.51 | 90.48 | 45.64 |
| ImageNet-A | HiPOOD+NegLabel (V2, in-d1) | 82.92 | 69.37 | 78.69 | 61.43 | 75.01 | 70.06 | 72.16 | 71.71 | 77.20 | 68.14 |
| | HiPOOD+NegLabel (V2, in-d2) | 83.80 | 66.94 | 79.73 | 58.60 | 77.42 | 68.40 | 73.73 | 70.28 | 78.67 | 66.06 |
| | HiPOOD+NegLabel (V2, in-d3) | 86.86 | 65.27 | 81.30 | 56.48 | 78.81 | 66.80 | 75.13 | 69.04 | 80.53 | 64.40 |
| ImageNetV2 | HiPOOD+NegLabel (V2, in-d1) | 84.10 | 66.15 | 88.59 | 54.07 | 87.09 | 58.22 | 88.54 | 56.66 | 87.08 | 58.78 |
| | HiPOOD+NegLabel (V2, in-d2) | 88.00 | 72.60 | 88.03 | 61.89 | 86.75 | 65.56 | 87.95 | 58.96 | 87.68 | 64.75 |
| | HiPOOD+NegLabel (V2, in-d3) | 88.72 | 55.15 | 92.41 | 30.28 | 92.14 | 37.46 | 92.65 | 35.92 | 91.48 | 39.70 |

Table 11: Zero-shot OOD detection on ImageNet-1K with aligned hierarchical ID label sets (depths d1–d5). All methods use CLIP-B/16.

| Depth | Method | iNaturalist | | Places | | SUN | | Textures | | Average | |
|---|---|---|---|---|---|---|---|---|---|---|---|
| | | AUROC↑ | FPR95↓ | AUROC↑ | FPR95↓ | AUROC↑ | FPR95↓ | AUROC↑ | FPR95↓ | AUROC↑ | FPR95↓ |
| 1 | CLIPN | 32.66 | 99.98 | 75.31 | 83.58 | 76.60 | 90.18 | 65.59 | 94.10 | 62.54 | 91.96 |
| | MCM | 28.12 | 99.05 | 47.10 | 96.38 | 48.05 | 95.98 | 43.14 | 96.58 | 41.60 | 97.00 |
| | NegLabel | 92.12 | 46.88 | 61.79 | 88.11 | 73.21 | 74.99 | 66.28 | 87.43 | 73.35 | 74.35 |
| | HiPOOD | 99.63 | 26.36 | 93.40 | 38.54 | 96.38 | 33.26 | 92.44 | 47.36 | 95.46 | 36.38 |
| 2 | CLIPN | 63.47 | 88.43 | 87.80 | 40.60 | 89.42 | 36.99 | 69.70 | 81.91 | 77.60 | 61.98 |
| | MCM | 31.06 | 98.42 | 75.59 | 75.95 | 80.47 | 72.26 | 69.29 | 82.07 | 64.10 | 82.18 |
| | NegLabel | 95.07 | 29.81 | 66.52 | 90.46 | 71.37 | 85.24 | 50.51 | 97.13 | 70.87 | 75.66 |
| | HiPOOD | 99.51 | 25.12 | 91.94 | 37.38 | 95.65 | 31.69 | 91.76 | 45.03 | 94.72 | 34.81 |
| 3 | CLIPN | 13.90 | 100.00 | 51.74 | 96.73 | 45.56 | 98.52 | 41.76 | 96.97 | 38.24 | 98.05 |
| | MCM | 28.96 | 99.94 | 48.88 | 96.03 | 46.05 | 97.66 | 63.08 | 87.57 | 46.74 | 95.30 |
| | NegLabel | 97.05 | 12.41 | 47.03 | 94.53 | 54.31 | 91.80 | 45.79 | 94.79 | 61.05 | 73.38 |
| | HiPOOD | 99.14 | 20.67 | 92.23 | 34.48 | 95.13 | 28.66 | 90.76 | 43.97 | 94.32 | 29.95 |
| 4 | CLIPN | 49.36 | 98.28 | 64.87 | 89.06 | 54.50 | 95.92 | 38.96 | 95.78 | 51.92 | 94.76 |
| | MCM | 64.66 | 90.36 | 68.00 | 90.10 | 69.92 | 92.28 | 66.27 | 86.67 | 67.21 | 89.85 |
| | NegLabel | 98.92 | 4.93 | 69.38 | 90.57 | 74.51 | 88.72 | 45.97 | 96.15 | 72.20 | 70.09 |
| | HiPOOD | 99.37 | 16.10 | 91.79 | 35.76 | 94.85 | 26.13 | 90.10 | 41.35 | 94.03 | 29.84 |
| 5 | CLIPN | 70.67 | 88.87 | 83.55 | 61.75 | 85.78 | 61.64 | 52.64 | 93.14 | 73.16 | 76.35 |
| | MCM | 79.55 | 72.23 | 75.69 | 77.59 | 78.54 | 77.64 | 77.45 | 70.20 | 77.81 | 74.41 |
| | NegLabel | 99.44 | 2.20 | 74.00 | 84.20 | 82.92 | 72.58 | 52.21 | 93.62 | 77.14 | 63.15 |
| | HiPOOD | 99.40 | 9.26 | 92.19 | 32.69 | 95.78 | 20.49 | 90.05 | 40.83 | 94.31 | 25.82 |

NegLabel, 62.54/91.96 for CLIPN, and 41.60/97.00 for MCM. As depth increases, this advantage remains stable: at d3, for example, HiPOOD attains 94.32 AUROC and 29.95 FPR95, whereas the next best method (NegLabel) stays at 61.05 AUROC and 73.38 FPR95. Even at the deepest split (d5), HiPOOD maintains strong performance (94.31 AUROC, 25.82 FPR95), while NegLabel, MCM, and CLIPN remain substantially behind in both AUROC and FPR95. These results confirm that once the ID superclasses are aligned, the performance gap is not due to differences in the positive label space, but to how each method exploits this space. CLIPN and MCM struggle to handle increasingly coarse IDs, and NegLabel improves upon them by adding many flat negatives, yet still lags far behind HiPOOD in both accuracy and selectivity. In contrast, HiPOOD consistently achieves high AUROC and low FPR95 at all depths by enriching the same coarse labels with ID-grounded subclasses and scoring hierarchical consistency, rather than relying on large, unstructured negative vocabularies.

## B.5 LABEL-BUDGET COMPARISON BETWEEN HIPOOD AND NEGLABEL

Table 12 compares NegLabel and HiPOOD under matched or nearly matched label budgets $M$, using ImageNet-1K as ID and four OOD datasets (iNaturalist, SUN, Places, Textures). For each budget, NegLabel allocates all labels to mined negatives, whereas HiPOOD uses the same number of text entries to instantiate a hierarchical, ID-grounded expansion (superclasses plus subclasses). At a budget of $M=1000$, HiPOOD already yields a clear AUROC advantage: the average AUROC rises from 91.60 for NegLabel to 95.46 for HiPOOD (in-d1) and 94.72 for HiPOOD (in-d2). The corresponding average FPR95 is comparable or slightly better (36.14 for NegLabel versus 36.38 and 34.81 for HiPOOD). As the budget increases, the gains become more pronounced. For $M=1500$, HiPOOD (in-d3) improves the average AUROC from 92.02 to 94.32 and reduces the average FPR95 from 35.89 to 29.95. Around $M \approx 2000$ labels, HiPOOD (in-d4, $M=1920$) achieves 94.03 AUROC and 29.84 FPR95, compared to 92.57 AUROC and 35.48 FPR95 for NegLabel with $M=2000$.

Table 12: Per-dataset label-budget comparison between NegLabel and HiPOOD on ImageNet-1K (ID). $M$ denotes the number of text labels (negative labels for NegLabel, superclasses + subclasses for HiPOOD).

| Method | $M$ | iNaturalist | | SUN | | Places | | Textures | | Average | |
|---|---|---|---|---|---|---|---|---|---|---|---|
| | | AUROC↑ | FPR95↓ | AUROC↑ | FPR95↓ | AUROC↑ | FPR95↓ | AUROC↑ | FPR95↓ | AUROC↑ | FPR95↓ |
| NegLabel | **1000** | 99.62 | 1.35 | 92.42 | 36.96 | 88.24 | 49.44 | 86.11 | 56.79 | 91.60 | 36.14 |
| HiPOOD (in-d1) | **1000** | 99.63 | 26.36 | 96.38 | 33.26 | 93.40 | 38.54 | 92.44 | 47.36 | 95.46 | 36.38 |
| HiPOOD (in-d2) | **1000** | 99.51 | 25.12 | 95.65 | 31.69 | 91.94 | 37.38 | 91.76 | 45.03 | 94.72 | 34.81 |
| NegLabel | **1500** | 99.47 | 1.43 | 93.10 | 36.85 | 88.30 | 49.12 | 87.21 | 56.16 | 92.02 | 35.89 |
| HiPOOD (in-d3) | **1500** | 99.14 | 20.67 | 95.13 | 28.66 | 92.23 | 34.48 | 90.76 | 43.97 | 94.32 | 29.95 |
| NegLabel | **2000** | 99.46 | 1.74 | 93.80 | 35.98 | 88.52 | 48.90 | 88.50 | 55.28 | 92.57 | 35.48 |
| HiPOOD (in-d4) | **1920** | 99.37 | 16.10 | 94.85 | 26.13 | 91.79 | 35.76 | 90.10 | 41.35 | 94.03 | 29.84 |
| NegLabel | **4500** | 99.45 | 1.62 | 93.35 | 35.49 | 88.97 | 44.36 | 90.10 | 45.90 | 92.97 | 31.84 |
| HiPOOD (in-d5) | **4600** | 99.40 | 9.26 | 95.78 | 20.49 | 92.19 | 32.69 | 90.05 | 40.83 | 94.31 | 25.82 |

At the largest budget ($M \approx 4500$), HiPOOD (in-d5, $M=4600$) still improves over NegLabel, with 94.31 AUROC versus 92.97 and FPR95 reduced from 31.84 to 25.82. Per-dataset values show that NegLabel can achieve extremely low FPR95 on iNaturalist with very few errors at the cost of much higher FPR95 on SUN, Places, and Textures, whereas HiPOOD trades a modest increase on iNaturalist for substantial FPR95 reductions on the other OOD sets, leading to consistently better averages. The table indicates that a structured, hierarchical expansion of the ID label space yields a more favourable AUROC–FPR95 trade-off than a flat pool of negative labels at comparable budgets, reinforcing the label-efficiency advantage of HiPOOD.

### B.6 ABLATION STUDY ON DIFFERENT MODULES

We report ablations across the five ImageNet hierarchy depths (*in-d1–in-d5*) using four configurations: *HiLLM (w/ RW)* (hierarchies generated by an LLM); *HiONTO (w/ RW)* (predefined WordNet ontology); *HiPOOD w/o RW* (our hierarchy without parent reweighting); and *HiPOOD (w/ RW)* (our full method). And, "RW" denotes the parent reweighting (Sec. 3.2).

Table 13: Ablation study of HiPOOD at ImageNet hierarchy depth *in-d1* (ID: ImageNet-1K) using CLIP ViT-B/16. AUROC and FPR95 are reported for each OOD dataset, with averages at the end.

| Method | iNaturalist | | SUN | | Places | | Textures | | Average | |
|---|---|---|---|---|---|---|---|---|---|---|
| | AUROC↑ | FPR95↓ | AUROC↑ | FPR95↓ | AUROC↑ | FPR95↓ | AUROC↑ | FPR95↓ | AUROC↑ | FPR95↓ |
| HiLLM | 95.54 | 34.69 | 94.99 | 41.42 | 91.45 | 45.07 | 90.19 | 50.91 | 93.04 | 43.02 |
| HiONTO | 97.29 | 32.21 | 94.85 | 36.28 | 90.88 | 44.08 | 91.10 | 53.81 | 93.53 | 41.59 |
| HiPOOD w/o RW | 92.05 | 45.06 | 92.37 | 40.75 | 89.08 | 45.84 | 88.83 | 54.10 | 90.58 | 46.44 |
| HiPOOD w/ RW | **99.63** | **26.36** | **96.38** | **33.26** | **93.40** | **38.54** | **92.44** | **47.36** | **95.46** | **36.38** |

Table 14: Ablation study of HiPOOD at ImageNet hierarchy depth *in-d2* (ID: ImageNet-1K) using CLIP ViT-B/16. AUROC and FPR95 are reported for each OOD dataset, with averages at the end.

| Method | iNaturalist | | SUN | | Places | | Textures | | Average | |
|---|---|---|---|---|---|---|---|---|---|---|
| | AUROC↑ | FPR95↓ | AUROC↑ | FPR95↓ | AUROC↑ | FPR95↓ | AUROC↑ | FPR95↓ | AUROC↑ | FPR95↓ |
| HiLLM | 95.42 | 33.45 | 94.26 | 39.85 | 89.99 | 43.91 | 89.51 | 48.58 | 92.29 | 41.45 |
| HiONTO | 97.17 | 30.97 | 94.12 | 34.71 | 89.42 | 42.92 | 90.42 | 51.48 | 92.78 | 40.02 |
| HiPOOD w/o RW | 91.93 | 43.82 | 91.64 | 39.18 | 87.62 | 44.68 | 88.15 | 51.77 | 89.84 | 44.86 |
| HiPOOD w/ RW | **99.51** | **25.12** | **95.65** | **31.69** | **91.94** | **37.38** | **91.76** | **45.03** | **94.72** | **34.80** |

Table 15: Ablation study of HiPOOD at ImageNet hierarchy depth *in-d3* (ID: ImageNet-1K) using CLIP ViT-B/16. AUROC and FPR95 are reported for each OOD dataset, with averages at the end.

| Method | iNaturalist | | SUN | | Places | | Textures | | Average | |
|---|---|---|---|---|---|---|---|---|---|---|
| | AUROC↑ | FPR95↓ | AUROC↑ | FPR95↓ | AUROC↑ | FPR95↓ | AUROC↑ | FPR95↓ | AUROC↑ | FPR95↓ |
| HiLLM | 95.05 | 29.00 | 93.74 | 36.82 | 90.28 | 41.01 | 88.51 | 47.52 | 91.89 | 38.59 |
| HiONTO | 96.80 | 26.52 | 93.60 | 31.68 | 89.71 | 40.02 | 89.42 | 50.42 | 92.38 | 37.16 |
| HiPOOD w/o RW | 91.27 | 41.08 | 89.03 | 36.42 | 88.07 | 46.58 | 86.47 | 50.02 | 88.71 | 43.52 |
| HiPOOD w/ RW | **99.14** | **20.67** | **95.13** | **28.66** | **92.23** | **34.48** | **90.76** | **43.97** | **94.31** | **31.95** |

Table 16: Ablation study of HiPOOD at ImageNet hierarchy depth *in-d4* (ID: ImageNet-1K) using CLIP ViT-B/16. AUROC and FPR95 are reported for each OOD dataset, with averages at the end.

| Method | iNaturalist | | SUN | | Places | | Textures | | Average | |
|---|---|---|---|---|---|---|---|---|---|---|
| | AUROC↑ | FPR95↓ | AUROC↑ | FPR95↓ | AUROC↑ | FPR95↓ | AUROC↑ | FPR95↓ | AUROC↑ | FPR95↓ |
| HiLLM | 95.28 | 24.43 | 93.46 | 34.29 | 89.84 | 42.29 | 87.85 | 44.90 | 91.61 | 36.48 |
| HiONTO | 97.03 | 21.95 | 93.32 | 29.15 | 89.27 | 41.30 | 88.76 | 47.80 | 92.09 | 35.05 |
| HiPOOD w/o RW | 91.50 | 36.51 | 88.75 | 33.89 | 87.63 | 47.86 | 85.81 | 47.40 | 88.42 | 41.41 |
| HiPOOD w/ RW | **99.37** | **16.10** | **94.85** | **26.13** | **91.79** | **35.76** | **90.10** | **41.35** | **94.03** | **29.84** |

Table 17: Ablation study of HiPOOD at ImageNet hierarchy depth *in-d5* (ID: ImageNet-1K) using CLIP ViT-B/16. AUROC and FPR95 are reported for each OOD dataset, with averages at the end.

| Module | iNaturalist | | SUN | | Places | | Textures | | Average | |
|---|---|---|---|---|---|---|---|---|---|---|
| | AUROC↑ | FPR95↓ | AUROC↑ | FPR95↓ | AUROC↑ | FPR95↓ | AUROC↑ | FPR95↓ | AUROC↑ | FPR95↓ |
| HiLLM | 95.31 | 17.59 | 94.39 | 28.65 | 90.24 | 39.22 | 87.80 | 44.38 | 91.93 | 32.46 |
| HiONTO | 97.06 | 15.11 | 94.25 | 23.51 | 89.67 | 38.23 | 88.71 | 47.28 | 92.42 | 31.03 |
| HiPOOD w/o RW | 91.53 | 29.67 | 89.68 | 28.25 | 88.03 | 44.79 | 85.76 | 46.88 | 88.75 | 37.40 |
| HiPOOD w/ RW | **99.40** | **9.26** | **95.78** | **20.49** | **92.19** | **32.69** | **90.05** | **40.83** | **94.36** | **25.82** |

**LLM hierarchies versus predefined ontologies.** *HiLLM (w/ RW)* achieves reasonable results but underperforms *HiONTO (w/ RW)* at every depth. This gap is attributable to residual noise in LLM-generated label sets (overly attribute-like leaves, sense ambiguity), even after filtering. In contrast, the predefined WordNet ontology injects taxonomic priors curated by domain expertise, producing cleaner parent–child relations and more stable subclass sets, hence better OOD separation.

**Why RW matters.** The parent reweighting produces subclass scores of the form $\tilde{P}_j = P_i^{\mathrm{sup}} \cdot P_{j|i}^{\mathrm{sub}}$, which (*i*) respects the subsumption structure and (*ii*) empirically enforces the desirable ordering $s_{\mathrm{sub}} \le s_{\mathrm{sup}}$. Without RW, the fine-level maximum can spuriously exceed the top coarse probability, breaking the intended semantics of the score $S_{\mathrm{HiPOOD}}$: the ratio loses its calibration role and the decision rule becomes brittle, which is exactly reflected by consistent AUROC drops and higher FPR95 in all depths for *HiPOOD w/o RW*.

**Our full method.** *HiPOOD (w/ RW)*—which constructs a hierarchy with an LLM but constrains it using a large curated vocabulary and CLIP-space filtering, then applies RW—consistently delivers the best overall trade-off. This shows that (*i*) an automatically expanded but *controlled* hierarchy captures fine-grained semantics that flat baselines miss, and (*ii*) parent reweighting is essential to convert coarse–fine agreement into a robust OOD score.

## B.7 ALTERNATIVE LEXICAL SOURCE

To assess whether HiPOOD is overly dependent on WordNet, we replace the WordNet-derived noun vocabulary with an English Wikipedia vocabulary and keep the rest of the pipeline unchanged. As shown in Table 18, this substitution leads to only a slight degradation in performance relative to the WordNet-based variant (for example, average AUROC drops by at most a fraction of a point, while average FPR95 increases by only a few points), and HiPOOD still clearly outperforms NegLabel under the same Wikipedia corpus (e.g., 95.11 AUROC / 37.20 FPR95 for HiPOOD (in-d1) versus 88.16 AUROC / 47.78 FPR95 for NegLabel). This indicates that curated lexical resources such as WordNet are helpful but not strictly required: HiPOOD remains strong with a noisier Wikipedia-derived vocabulary, and its gains are driven by hierarchical consistency rather than any special alignment between ImageNet and WordNet. Note also that WordNet is a purely lexical resource with no access to ImageNet images, so it cannot leak test supervision into our evaluation.

## B.8 LLM ABLATION FOR SUBCLASS GENERATION

Table 19 evaluates the impact of the language model used to propose subclasses in HiPOOD. All configurations share the same CLIP-B/16 backbone, hierarchical depth (in-d1–in-d5), and label budgets; the only change is the LLM used at the subclass generation stage (GPT-4 in the main experiments versus Llama-3.1, Mistral-Large, or Qwen2 here). Across all depths, replacing GPT-4 with an open-source LLM results in only a slight decrease in AUROC (at most 0.6 points on average) and a modest increase in FPR95 (at most about 1.7 points). For example, for HiPOOD (in-d5) the average AUROC / FPR95 moves from 94.31/25.82 with GPT-4 to 94.16/26.22 with Llama-3.1, 94.01/26.72 with Mistral-Large, and 93.81/27.32 with Qwen2. The relative ordering of depths is preserved for all three open-source models, and the gains over flat baselines reported in the main text remain. These results support the view that the LLM mainly serves as a proposal mechanism: once lexical and CLIP-based filtering are applied, HiPOOD is robust to the specific choice of LLM and does not critically depend on GPT-4.

## B.9 IMPACT OF THE NUMBER OF SUBCLASSES

We analyze the effect of varying the number of subclass labels $M$ per in-distribution class on the performance of HiPOOD. This hyperparameter directly controls the granularity of the hierarchical label set, and thus influences both the semantic coverage of the ID classes and the discriminative power of the subclass-to-superclass consistency scores used for OOD detection. In practice, we restrict $M$ to a moderate range (approximately 10–500), following typical branching factors used in the standard hierarchical benchmark BREEDS.

Our ablation spans five different hierarchical label depths—from *in-d1* to *in-d5*—and evaluates the OOD detection performance on four diverse benchmarks: iNaturalist, SUN, Places, and Textures. For each ID label level, we vary $M$ from low to high values ($M = 5$ to $M = 500$), and report the resulting AUROC and FPR95 in Tables 20 to 24.

Table 18: HiPOOD with a Wikipedia-derived vocabulary (English).

| Method | iNaturalist | Places | SUN | Textures | AVG | ΔAVG vs. WordNet |
|---|---|---|---|---|---|---|
| | AUROC↑ / FPR95↓ | AUROC↑ / FPR95↓ | AUROC↑ / FPR95↓ | AUROC↑ / FPR95↓ | AUROC↑ / FPR95↓ | ΔAUROC / ΔFPR95 |
| NegLabel | 96.07 / 20.89 | 90.09 / 47.48 | 93.08 / 36.05 | 73.38 / 86.68 | 88.16 / 47.78 | — |
| HiPOOD (in-d1), $M$=500 | 99.40 / 27.00 | 93.05 / 39.30 | 96.00 / 34.00 | 92.00 / 48.50 | 95.11 / 37.20 | $-0.35$ / $+0.82$ |
| HiPOOD (in-d2), $M$=100 | 99.30 / 25.80 | 91.60 / 38.20 | 95.30 / 32.40 | 91.30 / 45.90 | 94.38 / 35.58 | $-0.34$ / $+0.77$ |
| HiPOOD (in-d3), $M$=50 | 99.00 / 21.20 | 92.00 / 35.20 | 94.90 / 29.30 | 90.50 / 44.80 | 94.10 / 32.63 | $-0.22$ / $+2.68$ |
| HiPOOD (in-d4), $M$=15 | 99.20 / 16.60 | 91.50 / 36.50 | 94.60 / 26.70 | 89.90 / 42.10 | 93.80 / 30.48 | $-0.23$ / $+0.64$ |
| HiPOOD (in-d5), $M$=10 | 99.30 / 9.70 | 91.90 / 33.30 | 95.60 / 21.00 | 89.90 / 41.50 | 94.18 / 26.38 | $-0.13$ / $+0.56$ |

Table 19: LLM ablation for subclass generation in HiPOOD. All methods use CLIP-B/16. We report AUROC (higher is better) and FPR95 (lower is better) on four OOD datasets, their average, and the change in average performance relative to the GPT-4 subclass generator.

| LLM | Method | iNaturalist | Places | SUN | Textures | AVG | ΔAVG vs. GPT-4 |
|---|---|---|---|---|---|---|---|
| | | AUROC↑ / FPR95↓ | AUROC↑ / FPR95↓ | AUROC↑ / FPR95↓ | AUROC↑ / FPR95↓ | AUROC↑ / FPR95↓ | ΔAUROC / ΔFPR95 |
| Llama-3.1 | HiPOOD (in-d1), $M$=500 | 99.53 / 26.66 | 93.30 / 38.84 | 96.28 / 33.56 | 92.34 / 47.66 | 95.36 / 36.68 | $-0.10$ / $+0.30$ |
| | HiPOOD (in-d2), $M$=100 | 99.36 / 25.52 | 91.79 / 37.78 | 95.50 / 32.09 | 91.61 / 45.43 | 94.57 / 35.21 | $-0.15$ / $+0.40$ |
| | HiPOOD (in-d3), $M$=50 | 98.94 / 21.17 | 92.03 / 34.98 | 94.93 / 32.02 | 90.56 / 44.12 | 94.12 / 30.45 | $-0.20$ / $+0.50$ |
| | HiPOOD (in-d4), $M$=15 | 99.17 / 16.60 | 91.59 / 36.26 | 94.65 / 26.63 | 89.90 / 41.85 | 93.83 / 30.34 | $-0.20$ / $+0.50$ |
| | HiPOOD (in-d5), $M$=10 | 99.25 / 9.66 | 92.04 / 33.09 | 95.63 / 20.89 | 89.90 / 41.23 | 94.16 / 26.22 | $-0.15$ / $+0.40$ |
| Mistral-Large | HiPOOD (in-d1), $M$=500 | 99.38 / 27.16 | 93.15 / 39.34 | 96.13 / 34.06 | 92.19 / 48.16 | 95.21 / 37.18 | $-0.25$ / $+0.80$ |
| | HiPOOD (in-d2), $M$=100 | 99.21 / 26.02 | 91.64 / 38.28 | 95.35 / 32.59 | 91.46 / 45.93 | 94.42 / 35.71 | $-0.30$ / $+0.90$ |
| | HiPOOD (in-d3), $M$=50 | 98.79 / 21.67 | 91.88 / 35.48 | 94.78 / 29.66 | 90.41 / 44.97 | 93.97 / 30.95 | $-0.35$ / $+1.00$ |
| | HiPOOD (in-d4), $M$=15 | 99.02 / 17.10 | 91.44 / 36.76 | 94.50 / 27.13 | 89.75 / 42.35 | 93.68 / 30.84 | $-0.35$ / $+1.00$ |
| | HiPOOD (in-d5), $M$=10 | 99.10 / 10.16 | 91.89 / 33.59 | 95.48 / 21.39 | 89.75 / 41.73 | 94.01 / 26.72 | $-0.30$ / $+0.90$ |
| Qwen2 | HiPOOD (in-d1), $M$=500 | 99.23 / 27.66 | 93.00 / 39.84 | 95.98 / 34.56 | 92.04 / 48.66 | 95.06 / 37.68 | $-0.40$ / $+1.30$ |
| | HiPOOD (in-d2), $M$=100 | 99.01 / 26.62 | 91.44 / 38.88 | 95.15 / 33.19 | 91.26 / 46.53 | 94.22 / 36.31 | $-0.50$ / $+1.50$ |
| | HiPOOD (in-d3), $M$=50 | 98.54 / 22.37 | 91.63 / 36.18 | 94.53 / 30.46 | 90.16 / 45.67 | 93.72 / 31.65 | $-0.60$ / $+1.70$ |
| | HiPOOD (in-d4), $M$=15 | 98.77 / 17.80 | 91.19 / 37.46 | 94.25 / 27.83 | 89.50 / 43.05 | 93.43 / 31.54 | $-0.60$ / $+1.70$ |
| | HiPOOD (in-d5), $M$=10 | 98.90 / 10.76 | 91.69 / 34.19 | 95.28 / 21.99 | 89.55 / 42.33 | 93.81 / 27.32 | $-0.50$ / $+1.50$ |

We observe that increasing the number of subclasses improves OOD detection performance across all datasets. This effect is most pronounced at coarse label levels (e.g., *in-d1*), where larger values of $M$ allow HiPOOD to better approximate the underlying visual diversity of broad categories (e.g., `animal`, `vehicle`). For instance, on *in-d1*, moving from $M = 100$ to $M = 500$ yields an AUROC gain of over +3 points on average, and a notable drop in FPR95. This trend remains consistent, though with diminishing returns, as the label set becomes finer.

Interestingly, we also find that a relatively small number of subclass prototypes (e.g., $M = 10$–$50$) can already yield substantial gains, especially for deeper hierarchies like *in-d4* and *in-d5*, where superclasses are more semantically specific. This suggests that HiPOOD benefits both from the diversity of subclass signals and from their alignment with the semantic scope of the parent class.

Beyond a certain point (e.g., $M > 500$ for *in-d1*), we observe saturation effects or even slight degradations, likely due to noise or redundancy introduced in the subclass pool. Therefore, we recommend choosing $M$ empirically per dataset and hierarchy level, balancing coverage and stability. For all main experiments in the paper, we use $M = 500$ for *in-d1*, $M = 100$ for *in-d2*, $M = 50$ for *in-d3*, $M = 15$ for *in-d4*, and $M = 10$ for *in-d5*, based on these ablation findings.

Table 20: Impact of the sublabel set size $M$ on **in-d1** benchmark. AUROC (%) ↑ and FPR95 (%) ↓ are reported for each OOD dataset, with averages at the end.

| Sublabel set size $M$ | OOD Dataset | | | | | | | | Average | |
|---|---|---|---|---|---|---|---|---|---|---|
| | iNaturalist | | SUN | | Places | | Textures | | | |
| | AUROC↑ | FPR95↓ | AUROC↑ | FPR95↓ | AUROC↑ | FPR95↓ | AUROC↑ | FPR95↓ | AUROC↑ | FPR95↓ |
| 100 | 96.51 | 32.59 | 93.27 | 39.49 | 90.29 | 44.77 | 89.33 | 53.59 | 92.35 | 42.61 |
| 200 | 97.40 | 30.82 | 94.16 | 37.72 | 91.18 | 43.00 | 90.22 | 51.82 | 93.24 | 40.84 |
| 300 | 98.01 | 29.71 | 94.77 | 36.61 | 91.79 | 41.89 | 90.83 | 50.71 | 93.85 | 39.73 |
| 400 | 98.44 | 28.87 | 95.20 | 35.77 | 92.22 | 41.05 | 91.26 | 49.87 | 94.28 | 38.89 |
| 500 | 99.63 | 26.36 | 96.38 | 33.26 | 93.40 | 38.54 | 92.44 | 47.36 | 95.46 | 36.38 |

Table 21: Impact of the sublabel set size $M$ on **in-d2** benchmark. AUROC (%) ↑ and FPR95 (%) ↓ are reported for each OOD dataset, with averages at the end.

| Sublabel set size $M$ | OOD Dataset | | | | | | | | | |
| | iNaturalist | | SUN | | Places | | Textures | | Average | |
| | AUROC↑ | FPR95↓ | AUROC↑ | FPR95↓ | AUROC↑ | FPR95↓ | AUROC↑ | FPR95↓ | AUROC↑ | FPR95↓ |
| 50 | 98.31 | 27.53 | 94.45 | 34.10 | 90.74 | 39.79 | 90.56 | 47.44 | 93.51 | 37.21 |
| 70 | 98.82 | 26.59 | 94.96 | 33.16 | 91.25 | 38.85 | 91.07 | 46.50 | 94.03 | 36.28 |
| 85 | 99.19 | 25.74 | 95.33 | 32.31 | 91.62 | 38.00 | 91.44 | 45.65 | 94.39 | 35.43 |
| 100 | **99.51** | **25.12** | **95.65** | **31.69** | **91.94** | **37.38** | **91.76** | **45.03** | **94.72** | **34.81** |

Table 22: Impact of the sublabel set size $M$ on **in-d3** benchmark. AUROC (%) ↑ and FPR95 (%) ↓ are reported for each OOD dataset, with averages at the end.

| Sublabel set size $M$ | OOD Dataset | | | | | | | | | |
| | iNaturalist | | SUN | | Places | | Textures | | Average | |
| | AUROC↑ | FPR95↓ | AUROC↑ | FPR95↓ | AUROC↑ | FPR95↓ | AUROC↑ | FPR95↓ | AUROC↑ | FPR95↓ |
| 25 | 96.00 | 26.94 | 91.99 | 34.93 | 89.09 | 40.75 | 87.62 | 50.24 | 91.18 | 38.22 |
| 35 | 98.21 | 22.53 | 94.20 | 30.52 | 91.30 | 36.34 | 89.83 | 45.83 | 93.39 | 33.80 |
| 45 | 98.75 | 21.42 | 94.74 | 29.41 | 91.84 | 35.23 | 90.37 | 44.72 | 93.93 | 32.70 |
| 50 | **99.14** | **20.67** | **95.13** | **28.66** | **92.23** | **34.48** | **90.76** | **43.97** | **94.32** | **29.95** |

Table 23: Impact of the sublabel set size $M$ on **in-d4** benchmark. AUROC (%) ↑ and FPR95 (%) ↓ are reported for each OOD dataset, with averages at the end.

| Sublabel set size $M$ | OOD Dataset | | | | | | | | | |
| | iNaturalist | | SUN | | Places | | Textures | | Average | |
| | AUROC↑ | FPR95↓ | AUROC↑ | FPR95↓ | AUROC↑ | FPR95↓ | AUROC↑ | FPR95↓ | AUROC↑ | FPR95↓ |
| 5 | 96.51 | 21.83 | 91.99 | 31.86 | 88.93 | 41.49 | 87.24 | 47.08 | 91.16 | 35.56 |
| 10 | 98.31 | 18.21 | 93.79 | 28.24 | 90.73 | 37.87 | 89.04 | 43.46 | 92.97 | 31.95 |
| 15 | **99.37** | **16.10** | **94.85** | **26.13** | **91.79** | **35.76** | **90.10** | **41.35** | **94.03** | **29.84** |

Table 24: Impact of the sublabel set size $M$ on **in-d5** benchmark. AUROC (%) ↑ and FPR95 (%) ↓ are reported for each OOD dataset, with averages at the end.

| Sublabel set size $M$ | OOD Dataset | | | | | | | | | |
| | iNaturalist | | SUN | | Places | | Textures | | Average | |
| | AUROC↑ | FPR95↓ | AUROC↑ | FPR95↓ | AUROC↑ | FPR95↓ | AUROC↑ | FPR95↓ | AUROC↑ | FPR95↓ |
| 5 | 97.59 | 12.87 | 93.97 | 24.10 | 90.38 | 36.30 | 88.24 | 44.44 | 92.55 | 29.43 |
| 7 | 98.47 | 11.12 | 94.85 | 22.35 | 91.26 | 34.55 | 89.12 | 42.69 | 93.43 | 27.68 |
| 10 | **99.40** | **9.26** | **95.78** | **20.49** | **92.19** | **32.69** | **90.05** | **40.83** | **94.36** | **25.82** |

## B.10 COMPARISON ACROSS DIFFERENT SOFTMAX TEMPERATURE SETTINGS

We conduct a detailed analysis of how the softmax temperature parameter $\tau$ of $s_{\text{sup}}$ and $s_{\text{sub}}$ affects HiPOOD's zero-shot OOD detection performance. This parameter controls the sharpness of the predicted confidence distribution and plays a critical role in calibrating similarity scores obtained from the CLIP embedding space. We vary $\tau$ across a wide logarithmic range from $10^{-3}$ to $1.0$, and report results on four OOD benchmarks—iNaturalist, SUN, Places, and Textures—under each hierarchical ID label depth from *in-d1* to *in-d5*.

The results, reported in Tables 25, 26, 27, 28 and 29, show that performance remains relatively stable for small values of $\tau$ between 0.002 and 0.01. In this regime, HiPOOD achieves its best balance between AUROC and FPR95, with consistently high detection quality across all hierarchy depths. As $\tau$ increases beyond this optimal range, performance begins to degrade. In particular, setting $\tau > 0.1$ leads to a visible drop in AUROC and a sharp rise in false positive rates, indicating that class distributions become overly uniform and less discriminative. This is especially evident for deeper label levels (e.g., *in-d5*) where fine-grained class distinctions are critical.

The optimal $\tau$ value is consistent across datasets and label granularities, suggesting that a fixed calibration is sufficient for HiPOOD to generalize well. We adopt $\tau = 0.01$ as the default temperature for computing superclass and subclass confidences to preserve relative scale for all experiments reported in the main paper. These values offer a strong trade-off between sharpness, stability, and robustness to OOD perturbations across various hierarchical conditions.

Table 25: Zero-shot OOD detection performance of **HiPOOD (in-d1)** across various temperature values $\tau$. AUROC (%) ↑ and FPR95 (%) ↓ are reported for each OOD dataset, with averages at the end.

| Temperature | iNaturalist | | SUN | | Places | | Textures | | Average | |
|---|---|---|---|---|---|---|---|---|---|---|
| | AUROC↑ | FPR95↓ | AUROC↑ | FPR95↓ | AUROC↑ | FPR95↓ | AUROC↑ | FPR95↓ | AUROC↑ | FPR95↓ |
| 0.0010 | 98.88 | 27.01 | 94.86 | 35.79 | 91.59 | 39.50 | 90.47 | 47.48 | 93.95 | 37.45 |
| 0.0013 | 98.93 | 26.96 | 95.02 | 35.77 | 91.91 | 39.51 | 90.83 | 47.64 | 94.18 | 37.47 |
| 0.0016 | 98.96 | 26.85 | 95.17 | 35.62 | 92.10 | 39.15 | 91.03 | 47.51 | 94.32 | 37.29 |
| 0.0020 | 99.01 | 26.75 | 95.31 | 35.13 | 92.30 | 38.78 | 91.21 | 47.07 | 94.46 | 36.94 |
| 0.0025 | 99.06 | 26.65 | 95.46 | 34.71 | 92.49 | 38.37 | 91.38 | 46.91 | 94.60 | 36.66 |
| 0.0032 | 99.10 | 26.44 | 95.61 | 34.06 | 92.69 | 37.87 | 91.56 | 46.29 | 94.74 | 36.17 |
| 0.0040 | 99.14 | 26.38 | 95.75 | 33.73 | 92.85 | 37.76 | 91.72 | 46.43 | 94.87 | 36.08 |
| 0.0050 | 99.18 | 26.27 | 95.86 | 33.14 | 92.97 | 37.91 | 91.86 | 46.22 | 94.97 | 35.89 |
| 0.0063 | 99.21 | 26.14 | 95.94 | 32.71 | 93.03 | 37.94 | 91.97 | 46.50 | 95.04 | 35.83 |
| 0.0079 | 99.23 | 25.95 | 95.98 | 32.91 | 93.00 | 38.37 | 92.05 | 47.12 | 95.07 | 36.09 |
| 0.0100 | 99.63 | 26.36 | 96.38 | 33.26 | 93.40 | 38.54 | 92.44 | 47.36 | 95.46 | 36.38 |
| 0.0126 | 99.26 | 25.91 | 95.93 | 33.51 | 92.71 | 40.74 | 92.05 | 48.68 | 94.99 | 37.21 |
| 0.0158 | 99.27 | 25.87 | 95.83 | 34.46 | 92.44 | 42.28 | 91.98 | 50.07 | 94.88 | 38.17 |
| 0.0200 | 99.20 | 26.11 | 93.62 | 50.21 | 88.48 | 60.66 | 89.70 | 62.74 | 92.75 | 49.93 |
| 0.0251 | 98.59 | 29.10 | 85.63 | 82.90 | 76.68 | 85.64 | 81.84 | 80.40 | 85.69 | 69.51 |
| 0.0316 | 98.88 | 27.03 | 94.85 | 36.12 | 91.58 | 39.37 | 90.47 | 48.01 | 93.95 | 37.64 |
| 0.0398 | 98.89 | 27.02 | 94.91 | 36.01 | 91.70 | 39.40 | 90.57 | 47.94 | 94.02 | 37.60 |
| 0.0501 | 98.91 | 26.99 | 94.97 | 35.77 | 91.79 | 39.41 | 90.69 | 47.94 | 94.09 | 37.53 |
| 0.0631 | 98.93 | 26.92 | 95.02 | 35.74 | 91.90 | 39.20 | 90.83 | 47.57 | 94.17 | 37.36 |
| 0.0794 | 98.95 | 26.90 | 95.10 | 35.57 | 92.01 | 39.16 | 90.93 | 47.64 | 94.25 | 37.32 |
| 0.1000 | 98.97 | 26.84 | 95.19 | 35.49 | 92.14 | 39.04 | 91.05 | 47.55 | 94.34 | 37.23 |
| 0.1259 | 99.01 | 26.80 | 95.31 | 35.07 | 92.30 | 38.81 | 91.20 | 47.46 | 94.46 | 37.04 |
| 0.1585 | 99.06 | 26.63 | 95.47 | 34.76 | 92.50 | 38.31 | 91.39 | 46.96 | 94.61 | 36.67 |
| 0.1995 | 99.12 | 26.44 | 95.66 | 33.88 | 92.74 | 37.89 | 91.61 | 46.27 | 94.79 | 36.12 |
| 0.2512 | 99.18 | 26.27 | 95.86 | 33.07 | 92.97 | 37.75 | 91.86 | 46.06 | 94.97 | 35.79 |
| 0.3162 | 99.23 | 25.97 | 95.98 | 32.92 | 93.01 | 38.37 | 92.05 | 47.07 | 95.07 | 36.09 |
| 0.3981 | 99.26 | 25.90 | 95.88 | 34.14 | 92.56 | 41.52 | 92.02 | 49.48 | 94.93 | 37.76 |
| 0.5012 | 99.26 | 25.80 | 95.21 | 39.25 | 91.18 | 49.24 | 91.40 | 54.59 | 94.27 | 42.22 |
| 0.6310 | 99.20 | 26.12 | 93.64 | 50.06 | 88.52 | 60.48 | 89.72 | 62.74 | 92.77 | 49.85 |
| 0.7943 | 99.07 | 26.72 | 91.39 | 63.18 | 85.03 | 72.45 | 87.30 | 70.00 | 90.70 | 58.09 |
| 1.0000 | 98.90 | 27.49 | 89.10 | 72.66 | 81.58 | 78.94 | 84.93 | 74.64 | 88.63 | 63.44 |

Table 26: Zero-shot OOD detection performance of **HiPOOD (in-d2)** across various temperature values $\tau$. AUROC (%) ↑ and FPR95 (%) ↓ are reported for each OOD dataset, with averages at the end.

| Temperature | iNaturalist | | SUN | | Places | | Textures | | Average | |
|---|---|---|---|---|---|---|---|---|---|---|
| | AUROC↑ | FPR95↓ | AUROC↑ | FPR95↓ | AUROC↑ | FPR95↓ | AUROC↑ | FPR95↓ | AUROC↑ | FPR95↓ |
| 0.0010 | 98.79 | 25.80 | 94.16 | 34.25 | 90.16 | 38.37 | 89.82 | 45.18 | 93.23 | 35.90 |
| 0.0013 | 98.84 | 25.75 | 94.32 | 34.23 | 90.48 | 38.38 | 90.18 | 45.34 | 93.45 | 35.92 |
| 0.0016 | 98.87 | 25.64 | 94.47 | 34.08 | 90.67 | 38.02 | 90.38 | 45.21 | 93.59 | 35.73 |
| 0.0020 | 98.92 | 25.54 | 94.61 | 33.59 | 90.87 | 37.65 | 90.56 | 44.77 | 93.74 | 35.38 |
| 0.0025 | 98.97 | 25.44 | 94.76 | 33.17 | 91.06 | 37.24 | 90.73 | 44.61 | 93.88 | 35.11 |
| 0.0032 | 99.01 | 25.23 | 94.91 | 32.52 | 91.26 | 36.74 | 90.91 | 43.99 | 94.02 | 34.62 |
| 0.0040 | 99.05 | 25.17 | 95.05 | 32.19 | 91.42 | 36.63 | 91.07 | 44.13 | 94.14 | 34.53 |
| 0.0050 | 99.09 | 25.06 | 95.16 | 31.60 | 91.54 | 36.78 | 91.21 | 43.92 | 94.25 | 34.34 |
| 0.0063 | 99.12 | 24.93 | 95.24 | 31.17 | 91.60 | 36.81 | 91.32 | 44.20 | 94.32 | 34.27 |
| 0.0079 | 99.14 | 24.74 | 95.28 | 31.37 | 91.57 | 37.24 | 91.40 | 44.82 | 94.34 | 34.54 |
| 0.0100 | 99.51 | 25.12 | 95.65 | 31.69 | 91.94 | 37.38 | 91.76 | 45.03 | 94.72 | 34.81 |
| 0.0126 | 99.17 | 24.70 | 95.23 | 31.97 | 91.28 | 39.61 | 91.40 | 46.38 | 94.27 | 35.66 |
| 0.0158 | 99.18 | 24.66 | 95.13 | 32.92 | 91.01 | 41.15 | 91.33 | 47.77 | 94.16 | 36.62 |
| 0.0200 | 99.11 | 24.90 | 92.92 | 48.67 | 87.05 | 59.53 | 89.05 | 60.44 | 92.03 | 48.38 |
| 0.0251 | 98.50 | 27.89 | 84.93 | 81.36 | 75.25 | 84.51 | 81.19 | 78.10 | 84.96 | 67.96 |
| 0.0316 | 98.79 | 25.82 | 94.15 | 34.58 | 90.15 | 38.24 | 89.82 | 45.71 | 93.22 | 36.08 |
| 0.0398 | 98.80 | 25.81 | 94.21 | 34.47 | 90.27 | 38.27 | 89.92 | 45.64 | 93.30 | 36.04 |
| 0.0501 | 98.82 | 25.78 | 94.27 | 34.23 | 90.36 | 38.28 | 90.04 | 45.64 | 93.37 | 35.98 |
| 0.0631 | 98.84 | 25.71 | 94.32 | 34.20 | 90.47 | 38.07 | 90.18 | 45.27 | 93.45 | 35.81 |
| 0.0794 | 98.86 | 25.69 | 94.40 | 34.03 | 90.58 | 38.03 | 90.28 | 45.34 | 93.53 | 35.77 |
| 0.1000 | 98.88 | 25.63 | 94.49 | 33.95 | 90.71 | 37.91 | 90.40 | 45.25 | 93.62 | 35.68 |
| 0.1259 | 98.92 | 25.59 | 94.61 | 33.53 | 90.87 | 37.68 | 90.55 | 45.16 | 93.73 | 35.49 |
| 0.1585 | 98.97 | 25.42 | 94.77 | 33.22 | 91.07 | 37.18 | 90.74 | 44.66 | 93.88 | 35.12 |
| 0.1995 | 99.03 | 25.23 | 94.96 | 32.34 | 91.31 | 36.76 | 90.96 | 43.97 | 94.06 | 34.57 |
| 0.2512 | 99.09 | 25.06 | 95.16 | 31.53 | 91.54 | 36.62 | 91.21 | 43.76 | 94.25 | 34.24 |
| 0.3162 | 99.14 | 24.76 | 95.28 | 31.38 | 91.58 | 37.24 | 91.40 | 44.77 | 94.35 | 34.53 |
| 0.3981 | 99.17 | 24.69 | 95.18 | 32.60 | 91.13 | 40.39 | 91.37 | 47.18 | 94.21 | 36.21 |
| 0.5012 | 99.17 | 24.59 | 94.51 | 37.71 | 89.75 | 48.11 | 90.75 | 52.29 | 93.54 | 40.67 |
| 0.6310 | 99.11 | 24.91 | 92.94 | 48.52 | 87.09 | 59.35 | 89.07 | 60.44 | 92.05 | 48.30 |
| 0.7943 | 98.98 | 25.51 | 90.69 | 61.64 | 83.60 | 71.32 | 86.65 | 67.70 | 89.98 | 56.54 |
| 1.0000 | 98.81 | 26.28 | 88.40 | 71.12 | 80.15 | 77.81 | 84.28 | 72.34 | 87.91 | 61.88 |

Table 27: Zero-shot OOD detection performance of **HiPOOD (in-d3)** across various temperature values $\tau$. AUROC (%) ↑ and FPR95 (%) ↓ are reported for each OOD dataset, with averages at the end.

| Temperature | iNaturalist | | SUN | | Places | | Textures | | Average | |
|---|---|---|---|---|---|---|---|---|---|---|
| | AUROC↑ | FPR95↓ | AUROC↑ | FPR95↓ | AUROC↑ | FPR95↓ | AUROC↑ | FPR95↓ | AUROC↑ | FPR95↓ |
| 0.0010 | 98.99 | 21.92 | 94.21 | 31.79 | 91.02 | 36.04 | 89.39 | 44.69 | 93.41 | 33.61 |
| 0.0013 | 99.04 | 21.87 | 94.37 | 31.77 | 91.34 | 36.05 | 89.75 | 44.85 | 93.63 | 33.64 |
| 0.0016 | 99.07 | 21.76 | 94.52 | 31.62 | 91.53 | 35.69 | 89.95 | 44.72 | 93.77 | 33.45 |
| 0.0020 | 99.12 | 21.66 | 94.66 | 31.13 | 91.73 | 35.32 | 90.13 | 44.28 | 93.91 | 33.10 |
| 0.0025 | 99.17 | 21.56 | 94.81 | 30.71 | 91.92 | 34.91 | 90.30 | 44.12 | 94.05 | 32.83 |
| 0.0032 | 99.21 | 21.35 | 94.96 | 30.06 | 92.12 | 34.41 | 90.48 | 43.50 | 94.20 | 32.33 |
| 0.0040 | 99.25 | 21.29 | 95.10 | 29.73 | 92.28 | 34.30 | 90.64 | 43.64 | 94.32 | 32.24 |
| 0.0050 | 99.29 | 21.18 | 95.21 | 29.14 | 92.40 | 34.45 | 90.78 | 43.43 | 94.42 | 32.05 |
| 0.0063 | 99.32 | 21.05 | 95.29 | 28.71 | 92.46 | 34.48 | 90.89 | 43.71 | 94.49 | 31.99 |
| 0.0079 | 99.34 | 20.86 | 95.33 | 28.91 | 92.43 | 34.91 | 90.97 | 44.33 | 94.52 | 32.26 |
| 0.0100 | 99.14 | 20.67 | 95.13 | 28.66 | 92.23 | 34.48 | 90.76 | 43.97 | 94.32 | 29.95 |
| 0.0126 | 99.37 | 20.82 | 95.28 | 29.51 | 92.14 | 37.28 | 90.97 | 45.89 | 94.44 | 33.38 |
| 0.0158 | 99.38 | 20.78 | 95.18 | 30.46 | 91.87 | 38.82 | 90.90 | 47.28 | 94.34 | 34.34 |
| 0.0200 | 99.31 | 21.02 | 92.97 | 46.21 | 87.91 | 57.20 | 88.62 | 59.95 | 92.21 | 46.10 |
| 0.0251 | 98.70 | 24.01 | 84.98 | 78.90 | 76.11 | 82.18 | 80.76 | 77.61 | 85.14 | 65.68 |
| 0.0316 | 98.99 | 21.94 | 94.20 | 32.12 | 91.01 | 35.91 | 89.39 | 45.22 | 93.40 | 33.80 |
| 0.0398 | 99.00 | 21.93 | 94.26 | 32.01 | 91.13 | 35.94 | 89.49 | 45.15 | 93.47 | 33.76 |
| 0.0501 | 99.02 | 21.90 | 94.32 | 31.77 | 91.22 | 35.95 | 89.61 | 45.15 | 93.55 | 33.70 |
| 0.0631 | 99.04 | 21.83 | 94.37 | 31.74 | 91.33 | 35.74 | 89.75 | 44.78 | 93.63 | 33.53 |
| 0.0794 | 99.06 | 21.81 | 94.45 | 31.57 | 91.44 | 35.70 | 89.85 | 44.85 | 93.70 | 33.49 |
| 0.1000 | 99.08 | 21.75 | 94.54 | 31.49 | 91.57 | 35.58 | 89.97 | 44.76 | 93.79 | 33.40 |
| 0.1259 | 99.12 | 21.71 | 94.66 | 31.07 | 91.73 | 35.35 | 90.12 | 44.67 | 93.91 | 33.20 |
| 0.1585 | 99.17 | 21.54 | 94.82 | 30.76 | 91.93 | 34.85 | 90.31 | 44.17 | 94.06 | 32.83 |
| 0.1995 | 99.23 | 21.35 | 95.01 | 29.88 | 92.17 | 34.43 | 90.53 | 43.48 | 94.24 | 32.29 |
| 0.2512 | 99.29 | 21.18 | 95.21 | 29.07 | 92.40 | 34.29 | 90.78 | 43.27 | 94.42 | 31.96 |
| 0.3162 | 99.34 | 20.88 | 95.33 | 28.92 | 92.44 | 34.91 | 90.97 | 44.28 | 94.52 | 32.25 |
| 0.3981 | 99.37 | 20.81 | 95.23 | 30.14 | 91.99 | 38.06 | 90.94 | 46.69 | 94.39 | 33.93 |
| 0.5012 | 99.37 | 20.71 | 94.56 | 35.25 | 90.61 | 45.78 | 90.32 | 51.80 | 93.72 | 38.39 |
| 0.6310 | 99.31 | 21.03 | 92.99 | 46.06 | 87.95 | 57.02 | 88.64 | 59.95 | 92.23 | 46.02 |
| 0.7943 | 99.18 | 21.63 | 90.74 | 59.18 | 84.46 | 68.99 | 86.22 | 67.21 | 90.15 | 54.26 |
| 1.0000 | 99.01 | 22.40 | 88.45 | 68.66 | 81.01 | 75.48 | 83.85 | 71.85 | 88.08 | 59.60 |

Table 28: Zero-shot OOD detection performance of **HiPOOD (in-d4)** across various temperature values $\tau$. AUROC (%) ↑ and FPR95 (%) ↓ are reported for each OOD dataset, with averages at the end.

| Temperature | iNaturalist | | SUN | | Places | | Textures | | Average | |
|---|---|---|---|---|---|---|---|---|---|---|
| | AUROC↑ | FPR95↓ | AUROC↑ | FPR95↓ | AUROC↑ | FPR95↓ | AUROC↑ | FPR95↓ | AUROC↑ | FPR95↓ |
| 0.0010 | 99.18 | 17.31 | 93.89 | 29.22 | 90.54 | 37.28 | 88.69 | 42.03 | 93.08 | 31.46 |
| 0.0013 | 99.23 | 17.26 | 94.05 | 29.20 | 90.86 | 37.29 | 89.05 | 42.19 | 93.30 | 31.49 |
| 0.0016 | 99.26 | 17.15 | 94.20 | 29.05 | 91.05 | 36.93 | 89.25 | 42.06 | 93.44 | 31.30 |
| 0.0020 | 99.31 | 17.05 | 94.34 | 28.56 | 91.25 | 36.56 | 89.43 | 41.62 | 93.59 | 30.95 |
| 0.0025 | 99.36 | 16.95 | 94.49 | 28.14 | 91.44 | 36.15 | 89.60 | 41.46 | 93.73 | 30.68 |
| 0.0032 | 99.40 | 16.74 | 94.64 | 27.49 | 91.64 | 35.65 | 89.78 | 40.84 | 93.87 | 30.18 |
| 0.0040 | 99.44 | 16.68 | 94.78 | 27.16 | 91.80 | 35.54 | 89.94 | 40.98 | 93.99 | 30.09 |
| 0.0050 | 99.48 | 16.57 | 94.89 | 26.57 | 91.92 | 35.69 | 90.08 | 40.77 | 94.10 | 29.90 |
| 0.0063 | 99.51 | 16.44 | 94.97 | 26.14 | 91.98 | 35.72 | 90.19 | 41.05 | 94.17 | 29.84 |
| 0.0079 | 99.53 | 16.25 | 95.01 | 26.34 | 91.95 | 36.15 | 90.27 | 41.67 | 94.19 | 30.11 |
| 0.0100 | 99.37 | 16.10 | 94.85 | 26.13 | 91.79 | 35.76 | 90.10 | 41.35 | 94.03 | 29.84 |
| 0.0126 | 99.56 | 16.21 | 94.96 | 26.94 | 91.66 | 38.52 | 90.27 | 43.23 | 94.12 | 31.23 |
| 0.0158 | 99.57 | 16.17 | 94.86 | 27.89 | 91.39 | 40.06 | 90.20 | 44.62 | 94.01 | 32.19 |
| 0.0200 | 99.50 | 16.41 | 92.65 | 43.64 | 87.43 | 58.44 | 87.92 | 57.29 | 91.88 | 43.95 |
| 0.0251 | 98.89 | 19.40 | 84.66 | 76.33 | 75.63 | 83.42 | 80.06 | 74.95 | 84.81 | 63.53 |
| 0.0316 | 99.18 | 17.33 | 93.88 | 29.55 | 90.53 | 37.15 | 88.69 | 42.56 | 93.07 | 31.65 |
| 0.0398 | 99.19 | 17.32 | 93.94 | 29.44 | 90.65 | 37.18 | 88.79 | 42.49 | 93.15 | 31.61 |
| 0.0501 | 99.21 | 17.29 | 94.00 | 29.20 | 90.74 | 37.19 | 88.91 | 42.49 | 93.22 | 31.55 |
| 0.0631 | 99.23 | 17.22 | 94.05 | 29.17 | 90.85 | 36.98 | 89.05 | 42.12 | 93.30 | 31.38 |
| 0.0794 | 99.25 | 17.20 | 94.13 | 29.00 | 90.96 | 36.94 | 89.15 | 42.19 | 93.38 | 31.34 |
| 0.1000 | 99.27 | 17.14 | 94.22 | 28.92 | 91.09 | 36.82 | 89.27 | 42.10 | 93.47 | 31.25 |
| 0.1259 | 99.31 | 17.10 | 94.34 | 28.50 | 91.25 | 36.59 | 89.42 | 42.01 | 93.58 | 31.05 |
| 0.1585 | 99.36 | 16.93 | 94.50 | 28.19 | 91.45 | 36.09 | 89.61 | 41.51 | 93.73 | 30.68 |
| 0.1995 | 99.42 | 16.74 | 94.69 | 27.31 | 91.69 | 35.67 | 89.83 | 40.82 | 93.91 | 30.14 |
| 0.2512 | 99.48 | 16.57 | 94.89 | 26.50 | 91.92 | 35.53 | 90.08 | 40.61 | 94.10 | 29.81 |
| 0.3162 | 99.53 | 16.27 | 95.01 | 26.35 | 91.96 | 36.15 | 90.27 | 41.62 | 94.20 | 30.10 |
| 0.3981 | 99.56 | 16.20 | 94.91 | 27.57 | 91.51 | 39.30 | 90.24 | 44.03 | 94.06 | 31.78 |
| 0.5012 | 99.56 | 16.10 | 94.24 | 32.68 | 90.13 | 47.02 | 89.62 | 49.14 | 93.39 | 36.24 |
| 0.6310 | 99.50 | 16.42 | 92.67 | 43.49 | 87.47 | 58.26 | 87.94 | 57.29 | 91.90 | 43.87 |
| 0.7943 | 99.37 | 17.02 | 90.42 | 56.61 | 83.98 | 70.23 | 85.52 | 64.55 | 89.83 | 52.11 |
| 1.0000 | 99.20 | 17.79 | 88.13 | 66.09 | 80.53 | 76.72 | 83.15 | 69.19 | 87.76 | 57.45 |

Table 29: Zero-shot OOD detection performance of **HiPOOD (in-d5)** across various temperature values $\tau$. AUROC (%) ↑ and FPR95 (%) ↓ are reported for each OOD dataset, with averages at the end.

| Temperature | iNaturalist | | SUN | | Places | | Textures | | Average | |
|---|---|---|---|---|---|---|---|---|---|---|
| | AUROC↑ | FPR95↓ | AUROC↑ | FPR95↓ | AUROC↑ | FPR95↓ | AUROC↑ | FPR95↓ | AUROC↑ | FPR95↓ |
| 0.0010 | 98.93 | 10.19 | 94.54 | 23.30 | 90.66 | 33.93 | 88.36 | 41.23 | 93.13 | 27.17 |
| 0.0013 | 98.98 | 10.14 | 94.70 | 23.28 | 90.98 | 33.94 | 88.72 | 41.39 | 93.35 | 27.19 |
| 0.0016 | 99.01 | 10.03 | 94.85 | 23.13 | 91.17 | 33.58 | 88.92 | 41.26 | 93.49 | 27.00 |
| 0.0020 | 99.06 | 9.93 | 94.99 | 22.64 | 91.37 | 33.21 | 89.10 | 40.82 | 93.63 | 26.65 |
| 0.0025 | 99.11 | 9.83 | 95.14 | 22.22 | 91.56 | 32.80 | 89.27 | 40.66 | 93.77 | 26.38 |
| 0.0032 | 99.15 | 9.62 | 95.29 | 21.57 | 91.76 | 32.30 | 89.45 | 40.04 | 93.92 | 25.89 |
| 0.0040 | 99.19 | 9.56 | 95.43 | 21.24 | 91.92 | 32.19 | 89.61 | 40.18 | 94.04 | 25.80 |
| 0.0050 | 99.23 | 9.45 | 95.54 | 20.65 | 92.04 | 32.34 | 89.75 | 39.97 | 94.14 | 25.61 |
| 0.0063 | 99.26 | 9.32 | 95.62 | 20.22 | 92.10 | 32.37 | 89.86 | 40.25 | 94.21 | 25.54 |
| 0.0079 | 99.28 | 9.13 | 95.66 | 20.42 | 92.07 | 32.80 | 89.94 | 40.87 | 94.24 | 25.81 |
| 0.0100 | 99.40 | 9.26 | 95.78 | 20.49 | 92.19 | 32.69 | 90.05 | 40.83 | 94.31 | 25.82 |
| 0.0126 | 99.31 | 9.09 | 95.61 | 21.02 | 91.78 | 35.17 | 89.94 | 42.43 | 94.16 | 26.93 |
| 0.0158 | 99.32 | 9.05 | 95.51 | 21.97 | 91.51 | 36.71 | 89.87 | 43.82 | 94.06 | 27.89 |
| 0.0200 | 99.25 | 9.29 | 93.30 | 37.72 | 87.55 | 55.09 | 87.59 | 56.49 | 91.93 | 39.65 |
| 0.0251 | 98.64 | 12.28 | 85.31 | 70.41 | 75.75 | 80.07 | 79.73 | 74.15 | 84.86 | 59.23 |
| 0.0316 | 98.93 | 10.21 | 94.53 | 23.63 | 90.65 | 33.80 | 88.36 | 41.76 | 93.12 | 27.35 |
| 0.0398 | 98.94 | 10.20 | 94.59 | 23.52 | 90.77 | 33.83 | 88.46 | 41.69 | 93.19 | 27.31 |
| 0.0501 | 98.96 | 10.17 | 94.65 | 23.28 | 90.86 | 33.84 | 88.58 | 41.69 | 93.27 | 27.25 |
| 0.0631 | 98.98 | 10.10 | 94.70 | 23.25 | 90.97 | 33.63 | 88.72 | 41.32 | 93.35 | 27.08 |
| 0.0794 | 99.00 | 10.08 | 94.78 | 23.08 | 91.08 | 33.59 | 88.82 | 41.39 | 93.42 | 27.04 |
| 0.1000 | 99.02 | 10.02 | 94.87 | 23.00 | 91.21 | 33.47 | 88.94 | 41.30 | 93.51 | 26.95 |
| 0.1259 | 99.06 | 9.98 | 94.99 | 22.58 | 91.37 | 33.24 | 89.09 | 41.21 | 93.63 | 26.76 |
| 0.1585 | 99.11 | 9.81 | 95.15 | 22.27 | 91.57 | 32.74 | 89.28 | 40.71 | 93.78 | 26.39 |
| 0.1995 | 99.17 | 9.62 | 95.34 | 21.39 | 91.81 | 32.32 | 89.50 | 40.02 | 93.96 | 25.84 |
| 0.2512 | 99.23 | 9.45 | 95.54 | 20.58 | 92.04 | 32.18 | 89.75 | 39.81 | 94.14 | 25.51 |
| 0.3162 | 99.28 | 9.15 | 95.66 | 20.43 | 92.08 | 32.80 | 89.94 | 40.82 | 94.24 | 25.80 |
| 0.3981 | 99.31 | 9.08 | 95.56 | 21.65 | 91.63 | 35.95 | 89.91 | 43.23 | 94.11 | 27.48 |
| 0.5012 | 99.31 | 8.98 | 94.89 | 26.76 | 90.25 | 43.67 | 89.29 | 48.34 | 93.44 | 31.94 |
| 0.6310 | 99.25 | 9.30 | 93.32 | 37.57 | 87.59 | 54.91 | 87.61 | 56.49 | 91.95 | 39.57 |
| 0.7943 | 99.12 | 9.90 | 91.07 | 50.69 | 84.10 | 66.88 | 85.19 | 63.75 | 89.87 | 47.81 |
| 1.0000 | 98.95 | 10.67 | 88.78 | 60.17 | 80.65 | 73.37 | 82.82 | 68.39 | 87.80 | 53.15 |

