# OpenReview forum: "HiPOOD: Hierarchical Prompt-Aware Zero-Shot Out-of-Distribution Detection"
_ICLR.cc/2026/Conference — Submitted to ICLR 2026_

### Official Review · Reviewer_38av · 2025-10-29

**Soundness:** 3
**Presentation:** 3
**Contribution:** 3
**Rating:** 4
**Confidence:** 4

**Summary:**

This paper proposes HIPOOD, a method for zero-shot out-of-distribution (OOD) detection that leverages hierarchical structures of in-distribution (ID) labels. Specifically, the approach uses a large language model (LLM) to generate fine-grained subcategories for each ID label, thereby creating a semantic hierarchy. By utilizing both the ID labels and the newly derived fine-grained labels, the model achieves superior detection performance compared to existing methods.

**Strengths:**

- S1. The problem setting is interesting. While previous studies have used LLMs to generate OOD labels, this paper instead utilizes them to subdivide ID labels, which represents a novel perspective.

- S2. The explanations throughout the paper are clear, and the distinction from existing work is well articulated, making the contribution easy to understand.

- S3. The paper conducts a large number of experiments, and the ablation studies.

**Weaknesses:**

-  W1.In the results section, all comparison methods use the original ImageNet labels, while the proposed method uses a different set of labels. It would be better to align the labels across methods to ensure a fair comparison.

- W2.Within OOD detection, hard OOD detection has recently become an important focus, yet this paper does not report results for that setting. Since performance on common OOD detection is already quite high in general, it would be important to include and discuss hard OOD detection results as well [1]. It would be better to include the results with NINCO [2] and SSB-Hard [3].

-  W3. For the evaluation of ImageNet variants, the NegLabel method performs better. Therefore, the effectiveness of the proposed method is not fully demonstrated. It would be beneficial to examine whether combining the proposed approach with NegLabel improves performance.

-  W4. The method appears to be sensitive to hyperparameters, particularly to the relationship between depth and M. The optimal value of M seems to vary depending on the depth, and performance changes significantly with different M values. This sensitivity may make the method less practical or difficult to use in real-world scenarios.

[1] Noda+, A Benchmark and Evaluation for Real-World Out-of-Distribution Detection Using Vision-Language Models, ICIP2025

[2] Bitterwolf+, In or Out? Fixing ImageNet Out-of-Distribution Detection Evaluation, ICML2023

[3] Vaze+, Open-Set Recognition: a Good Closed-Set Classifier is All You Need?, ICLR2022

**Questions:**

- I would like to see the performance of the comparison methods reported when they use the same coarse labels as the proposed method.

- It would also be valuable to include results on hard OOD detection.

- When the domain differs, the NegLabel method performs better, which suggests that the effectiveness of the proposed approach is not fully demonstrated. It may be beneficial to explore combining the proposed method with NegLabel to further improve performance.

- Regarding the sensitivity to hyperparameters, it would strengthen the paper if there were a clear and convincing justification for why such sensitivity is acceptable.

---

> ### Author Response · Authors · 2025-11-24
> **Reply to Reviewer 38av**
>
> ### **4.1 Fairness of comparison: different ID label sets across methods**
>
> We thank the reviewer for pointing out that the original comparison could be biased by the fact that methods operate on flat ImageNet-1K labels, while HiPOOD relies on hierarchical splits (in-d1 to in-d5). We agree that a fair comparison should rely on *aligned* ID label sets and comparable label budgets.
>
> In the revised version, we therefore re-evaluate under the same hierarchy of *coarse* ID labels that HiPOOD uses as superclasses. For each depth d1-d5, we construct the corresponding ID label set (in-d1 to in-d5) and use this *same set of superclasses* as the positive label space for MCM, CLIPN, NegLabel, and HiPOOD. Each method then applies its own mechanism on top of these shared superclasses. This aligned setting is described in Appendix B.4, Table 11.
>
> We also explicitly control for the *total* label budget. Appendix B.5 (Table 12) summarizes this comparison. HiPOOD improves average AUROC by about $+1$–$4$ and reduces average FPR95 by about $4$–$6$ points across budgets. For example, at $M\approx1500$, NegLabel reaches $\sim92$ AUROC and $\sim36$ FPR95, while HiPOOD (in-d3) reaches $\sim94$ AUROC and $\sim30$ FPR95. At $M\approx4500$, HiPOOD (in-d5) still improves both metrics over NegLabel.
>
> These aligned-label experiments and label-budget analyses show that once (i) ID superclasses are shared and (ii) total label count is controlled, HiPOOD consistently offers a better AUROC–FPR95 trade-off through structured ID-grounded expansion rather than large flat negative vocabularies.
>
> ### **4.2 Missing evaluation on hard OOD benchmarks (NINCO, SSB-hard)**
>
> We add results on NINCO and SSB-hard using the same CLIP-B/16 backbone and zero-shot setting.
>
> Figure 4 summarizes these results. On NINCO, HiPOOD (in-d1) slightly improves AUROC and FPR95 over MCM and clearly surpasses NegLabel. On SSB-hard, HiPOOD (in-d5) achieves the best AUROC and FPR95 among all methods. Combined with domain-shift results on ImageNet-Sketch, ImageNet-A, and ImageNetV2 (Table 6, Figure 3), this shows that hierarchical consistency remains effective in hard near-OOD regimes where flat baselines struggle.
>
> ### **4.3 NegLabel vs. HiPOOD under domain shift; combining both scores**
>
> #### **Domain-shifted ID data**
>
> Section 4.4 and Table 6 compare methods when ID data come from Sketch, ImageNet-A, and ImageNetV2. Two patterns hold:
>
> 1. Increasing depth (in-d1 to in-d3) improves HiPOOD.
> 2. The hierarchical approach closes or surpasses flat baselines.
>
> On Sketch, in-d3 nearly matches NegLabel; on V2 it surpasses it; on A a gap remains, but improvements over MCM are large. HiPOOD remains competitive under domain shift despite using a compact ID-grounded expansion.
>
> #### **Combining hierarchical consistency with negative labels**
>
> Appendix B.2 defines two hybrid scores:
>
> * a **hierarchical negative ratio**
>   $S_{\mathrm{HiPOOD\text{-}NegLabel}}^{(1)}(\mathbf{x}) $,
>
>
> * a **linear combination**
>   $S_{\mathrm{HiPOOD\text{-}NegLabel}}^{(2)}(\mathbf{x}) = 2\lambda S_{\mathrm{HiPOOD}}(\mathbf{x}) + (1-\lambda) S_{\mathrm{NegLabel}}(\mathbf{x})$.
>
> Per-dataset results (Tables 9–10) show:
>
> * $S_{\mathrm{HiPOOD\text{-}NegLabel}}^{(1)}$ slightly reduces FPR95 with a small AUROC drop (~1–2 points).
> * $S_{\mathrm{HiPOOD\text{-}NegLabel}}^{(2)}$ behaves between the two base detectors and does not consistently dominate either.
>
> Thus the combination is optional; for simplicity and interpretability we keep $S_{\mathrm{HiPOOD}}(\mathbf{x})$ as the main score.
>
> ### **4.4 Sensitivity to $M$ and hierarchy depth**
>
> $M$ is the branching factor (subclasses per superclass) and depth controls hierarchical granularity. These are structural choices, not tuned hyperparameters.
>
> Appendix B.6 reports an ablation over $M$ across depths. Performance shows a broad plateau: degradation occurs only when $M$ is extremely small (no hierarchy) or extremely large (redundancy/noise). Within the selected range ($10$–$500$), HiPOOD is stable and the same $M$ is used across datasets.
>
> We clarify in the final version that both $M$ and depth come from simple, dataset-agnostic ranges and that HiPOOD exhibits robust behavior, as documented in Section 4 and Appendix B.

---

> > ### Comment · Reviewer_38av · 2025-11-27
> > **Response from Reviewer 38av**
> >
> > I appreciate the authors’ careful rebuttal. In particular, I am grateful that they added experiments on Hard OOD detection and reported the performance of existing methods using labels at the same depth as HIPOOD to ensure a fair comparison.
> >
> > Since my initial concerns have been resolved, I have increased my score accordingly.
> >
> > However, at the same time, I now realize that the performance gains of the proposed method are guaranteed only under very specific conditions. More specifically, the method is applicable only in settings where a hierarchical label structure exists (such as ImageNet) and where the user is given only super-class labels. The approach of HIPOOD appears to be effective exclusively within this problem setup.
> >
> > While the authors attempted to address this issue in Table 8 of the revised version, the performance on CUB-200, Food-101, and FGVC-Aircraft is already around 98–99%, leaving little room for meaningful comparison or discussion.
> >
> > Therefore, I still remain uncertain about the broader academic value of a method whose performance improvements are limited to such a niche problem setting.
> >
> > However, I do believe that the authors have demonstrated the effectiveness of their approach thoroughly within the proposed problem setting. For this reason, I am raising my score to 6.
> > I plan to discuss with the other reviewers and the AC about my remaining concern in the next phase.

---

> > ### Author Response · Authors · 2025-11-27
> >
> > We sincerely thank the reviewer for the positive update and for recognizing the effectiveness of HiPOOD within the proposed setting. We address the remaining concern regarding the scope of applicability.
> >
> > HiPOOD is not restricted to scenarios where only super-class labels are provided or where a predefined hierarchy exists. The core contribution is that ID labels admit a meaningful refinement into subcategories (coarse-to-fine structure), which we can *induce* from text. Our pipeline generates candidate subclasses from an LLM and then enforces grounding and de-duplication through lexical constraints and CLIP-based filtering (see Section 3.1 and Appendix B.6). This means HiPOOD can be applied to any dataset with textual class names whenever such coarse-fine refinements are semantically plausible, even if the dataset is originally presented as “flat”. For ID classes that are already very fine-grained or lack clear subcategories, LLMs often still propose useful attribute or condition-style variants; after filtering, we retain only visually grounded subclasses, and we can fall back to keeping the parent label as a pseudo-subclass and we suffix all retained candidates with the parent name when needed (see Appendix A.1).
> >
> > We agree that Table 8 (CUB-200, Food-101, FGVC-Aircraft) is near-saturated for all methods, leaving limited room for meaningful metric separation. HiPOOD still operates correctly because it builds its own semantic hierarchy rather than relying on an existing one.
> >
> > To evaluate gains in non-saturated regimes, the revised paper includes (i) hard OOD benchmarks (NINCO, SSB-hard; Appendix B.1) and (ii) depth-aligned comparisons where all baselines share the same coarse label sets as HiPOOD (Appendix B.4). In these settings, HiPOOD remains competitive or improves over strong baselines, indicating that the benefit is not an artifact of mismatched label spaces but comes from exploiting coarse–fine semantic consistency. We further support robustness via ablations that replace WordNet with Wikipedia and GPT-4 with open-source LLMs (Appendix B.7 and Appendix B.8), where performance degrades only moderately and the main trends are preserved.
> >
> > Beyond performance, HiPOOD provides an explicit and interpretable signal by decomposing detection into superclass confidence and best subclass agreement (Section 3.2), making rejections explainable (e.g., "coarse: yes / fine: no"), which is valuable when users naturally reason in categories and subcategories (since they are accustomed to hierarchical reasoning).
> >
> > In summary, while HiPOOD indeed excels in the super-class scenario used to motivate the problem; it applies whenever categories can be sensibly refined into subcategories, and the revised experiments (hard OOD, aligned-label evaluation, and vocabulary/LLM ablations) support that its gains are not confined to a narrow or artificial setup.

---

### Official Review · Reviewer_o7u9 · 2025-10-29

**Soundness:** 2
**Presentation:** 2
**Contribution:** 2
**Rating:** 4
**Confidence:** 4

**Summary:**

The paper proposes HiPOOD, a zero-shot OOD detection method that augments CLIP with LLM-generated hierarchical subclass labels. It constructs per-class fine-grained label sets via prompting and vocabulary filtering, then computes a coarse-to-fine consistency score to flag OOD samples. The method claims SOTA zero-shot OOD performance on standard benchmarks and improved interpretability.

**Strengths:**

- The method is zero-shot and training-free, requiring no ID images, fine-tuning, or auxiliary classifiers—only pre-trained CLIP, one-time LLM queries per class, and a static WordNet vocabulary—enabling instant deployment on resource-constrained devices.

- By targeting naturally hierarchical label spaces, HiPOOD achieves superior sensitivity to near-OOD anomalies via parent-conditional reweighting, correctly flagging fine-grained unknowns (e.g., wolf under "dog") where coarse confidence misleads MSP/MCM, leveraging VLM calibration and avoiding exhaustive negative label coverage.

**Weaknesses:**

- Novelty is limited as the core hierarchical framework—parent-conditional softmax, superclass reweighting, coarse-to-fine consistency—is a near-direct extension of CHiLS, with HiPOOD’s contribution reduced to LLM-based subclass regeneration plus percentile filtering, an engineering tweak rather than a conceptual advance; given ImageNet-1K’s existing WordNet hierarchy, regenerating it via GPT-4 may add noise without justification, failing to address any fundamental prior limitation.

- Heavy, unstable reliance on closed-source LLMs (GPT-4) undermines reproducibility and robustness, with no ablation across open-source models, prompt variations, or failure modes (e.g., generic/irrelevant outputs like "pet"), risking entire subclass tree corruption and performance collapse without controlled external dependencies.

- Computational and deployment overhead is drastically underestimated—for ImageNet-1K (N=1000, M=10), offline phase requires ~1,000 costly LLM calls plus potentially millions of CLIP text encodings over a ~80,000-noun WordNet vocabulary, while online inference demands ~11,000 cosine similarities and 1,000 softmaxes per image, orders of magnitude slower than MSP/energy scoring, with zero discussion of latency, cost, or scalability trade-offs.

- Evaluation is severely overstated and misleadingly narrow, claiming applicability to real-world hierarchical domains like medical imaging (SNOMED/UMLS) while testing exclusively on ImageNet-1K—a dataset with a pre-existing, clean WordNet hierarchy that creates implicit data leakage and an artificially favorable setting; the paper presents zero results on non-hierarchical datasets (CIFAR-10/100), domain-specific data (medical/satellite/e-commerce), and omits critical ablations on LLM quality, prompt design, percentile threshold, or vocabulary source.

**Questions:**

If the ID category itself is fine-grained, how can authors generate subcategories to accommodate the proposed method? Additionally, certain categories may not have subcategories, such as traffic lights.

---

> ### Author Response · Authors · 2025-11-24
> **Reply to Reviewer o7u9**
>
> ### **3.1 Novelty vs. CHiLS and “subclass regeneration”**
>
> We thank the reviewer for highlighting the connection to CHiLS. CHiLS is indeed an important inspiration, HiPOOD differs in several key aspects, beyond ontology regeneration.
>
> First, HiPOOD introduces a *zero-shot hierarchical detection score* built directly from CLIP similarities, without any training on ID images or prompt tuning. The hierarchy is not only used to improve classification, but to define an explicit *coarse-to-fine OOD score* that serves as the main detection signal.
>
> Second, HiPOOD uses a *parent-conditional consistency ratio* $S_{\mathrm{HiPOOD}}(\mathbf{x})$ that measures agreement between coarse and fine levels. To our knowledge, no prior work turns hierarchical *consistency itself* into a zero-shot OOD score built directly from VLM similarities.
>
> Third, the subclass induction pipeline is designed for OOD detection rather than WordNet regeneration: the LLM+vocabulary mechanism enriches the label space with visually grounded subclasses, filters out duplicates/near-duplicates, and can operate with or without a predefined ontology.
>
> ### **3.2 Reliance on GPT-4 and open-source LLM ablations**
>
> We agree that relying only on GPT-4 would raise reproducibility concerns. In the revised experiments, we therefore regenerate subclasses with several open-source LLMs (Llama-3.1, Mistral, Qwen) and re-run HiPOOD across all ImageNet-1K hierarchy depths (in-d1–in-d5) and OOD datasets (iNaturalist, SUN, Places, Textures).
>
> As reported in Appendix B.7 (Table 19), the average AUROC changes only slightly and the average FPR95 increases moderately compared to the GPT-4-based hierarchy; the relative ranking of hierarchy depths is preserved, and HiPOOD remains consistently better than flat baselines under comparable label budgets. This supports the claim that the LLM mainly acts as a *proposal mechanism*, while effective subclasses are largely determined by vocabulary constraints and CLIP-based filtering.
>
> ### **3.3 Computational overhead and scalability**
>
> All text embeddings for superclasses and subclasses are computed *offline* once and cached; no backpropagation nor LLM calls are needed at evaluation time. At *test time*, HiPOOD performs a single CLIP forward pass per image plus dot products between the image embedding and cached text embeddings:
>
> (i) coarse similarities over all $N$ superclasses, and
>
> (ii) fine similarities over subclasses of a small set of top-$K$ parents ($K \ll N$).
>
> The resulting complexity per image is
> $O(N + K \cdot M_{\text{parent}})$,
> comparable to other CLIP-based zero-shot methods and scaling linearly with the size of the hierarchical label set.
>
> ### **3.4 Evaluation breadth, WordNet bias, and domain-specific datasets**
>
> #### **Alternative lexical source (Wikipedia)**
>
> To test whether HiPOOD is overly tied to WordNet, we replace the WordNet-based vocabulary by a Wikipedia-derived one and re-run both HiPOOD and compare it also to NegLabel. As detailed in Appendix B.6 (Table 18), performance decreases moderately for both methods, but overall trends are preserved and HiPOOD retains a clear advantage under matched label budgets.
>
> #### **Non-hierarchical and domain-specific ID datasets**
>
> Beyond ImageNet-1K, we evaluate on fine-grained datasets such as CUB-200 [1], Food-101 [2], and FGVC-Aircraft [3]. Appendix B.2 reports inter-dataset OOD detection results. HiPOOD consistently improves over MCM and remains close to or competitive with NegLabel in AUROC and FPR95. On FGVC-Aircraft, NegLabel attains the best metrics, but HiPOOD significantly narrows the gap relative to MCM.
>
> These results indicate that hierarchical prompt-aware reasoning can help even when the label space is not explicitly structured as a taxonomy. WordNet is purely lexical; the additional Wikipedia and domain-specific experiments support that HiPOOD does not rely on any unfair advantage from it.
>
> [1] Wah et al., *The caltech-ucsd birds-200-2011 dataset.* 2011.
>
> [2] Bossard et al., *Food-101–mining discriminative components with random forests.* In ECCV, 2014.
>
> [3] Maji et al., *Fine-Grained Visual Classification of Aircraft.* 2013.
>
> ### **3.5 Fine-grained or “leaf-like” ID classes**
>
> For ID classes already very fine-grained or lacking natural subcategories (e.g., *traffic light*), LLMs often still propose meaningful attribute- or condition-style variants (e.g., color, state). Vocabulary and CLIP-based filtering discards uninformative candidates, keeping only visually grounded subclasses.
>
> When neither vocabulary nor CLIP filtering yields reliable subclasses, we can apply a simple back-off strategy: keep the parent label as a pseudo-subclass and constrain candidates to remain explicitly tied to the parent. In such cases, the hierarchy remains shallow and the HiPOOD score reduces to a coarse-level score with mild regularization. These cases are rare and have limited impact on aggregate metrics. We summarize this back-off mechanism in Appendix A.1.

---

### Official Review · Reviewer_yEQc · 2025-10-30

**Soundness:** 2
**Presentation:** 2
**Contribution:** 2
**Rating:** 2
**Confidence:** 4

**Summary:**

This paper proposes a zero-shot out-of-distribution detection method named HiPOOD, which enhances the representation of known categories by constructing a semantic class hierarchy and uses the CLIP model to conduct coarse-grained and fine-grained consistency evaluation at the hierarchy, thereby identifying OOD samples.

**Strengths:**

1. The paper is well written.
2. Experiments are adequate, verifying the advantage of the proposed HiPOOD.

**Weaknesses:**

1. The authors argue that "There will always be unseen unknowns that are not represented by any pre-collected negative label". However, such a issue also exists when using hierarchy, where unseen unknowns could always appear out of the admitted hierarchy.
2. Is the hierarchy consistency on the semantic relation kept the same as the CLIP-based embedding similarity? And corarse-level classification may not be that accurate for CLIP-based prompt, since the pre-training data would pair the images with class names from various granularity.
3. The idea of introducing hierarchy into OOD detection has been widely used in related methods, leading to the method lack of novelty.
4. Comparison to a Stronger "Hierarchical" Baseline: While compared to many flat-label methods, the paper could be strengthened by a direct comparison to a simpler hierarchical baseline.
5. The performance hinges on the quality of the generated subclasses. The ablation shows that a predefined ontology can outperform the LLM-generated hierarchy. This raises questions about the method's robustness across diverse domains, especially where high-quality lexical resources like WordNet are unavailable.

**Questions:**

See the weakness.

---

> ### Author Response · Authors · 2025-11-24
> **Reply to reviewer yEQc (part 1)**
>
> ### **2.1 Clarification on unseen unknowns and hierarchy coverage**
>
> We thank the reviewer for raising the important point that a semantic hierarchy cannot cover all unseen unknowns. We fully agree, and we emphasize that HiPOOD is not designed to enumerate the unknown space. Instead, it focuses on detecting *inconsistencies between coarse- and fine-level predictions* within a given semantic hierarchy: intuitively, images that are reasonably compatible with a coarse concept (e.g., “dog”) but fail to align with any of its plausible subclasses under our parent-conditional formulation.
>
> In that sense, HiPOOD plays a different role from negative-label methods. Rather than carving out “not ID’’ via many explicit negative anchors, it flags samples that break hierarchical consistency. This mechanism does not require the hierarchy to span all unknown classes; it only requires that many OOD examples induce a mismatch between coarse and fine predictions, which we show empirically across several domain-shifted ImageNet variants.
>
> Regarding coverage vs. label budget, our experiments systematically match the *total* number of text labels between NegLabel and HiPOOD (ID labels plus negatives vs. ID superclasses plus subclasses). As summarized in Table 12, HiPOOD consistently improves average AUROC (typically by $+1$–$4$ points) and reduces FPR95 (by $4$–$6$ points) over NegLabel under comparable budgets. This supports our claim that HiPOOD relies on *hierarchical consistency within a compact, structured label set*, rather than on exhaustive modeling of the unknown space.
>
>
> ### **2.2 Alignment between hierarchy semantics and CLIP embeddings**
>
> We appreciate the concern that CLIP’s geometry does not perfectly mirror linguistic hierarchies. HiPOOD explicitly exploits the *contrast* between coarse and fine predictions.
>
> Empirically, prior work (e.g., Bianchi et al., 2024; Kim & Ji, 2024) and our own analysis indicate that CLIP tends to represent coarse-grained concepts (such as “dog”, “bird”, “car”) more robustly than fine-grained subtypes, and that mixed-granularity captions can blur the relationship between parent and child labels. HiPOOD turns this behaviour into a signal: for ID images we typically observe coherent coarse–fine paths (a confident superclass and at least one consistent subclass), while OOD images often exhibit the opposite pattern (weak or unstable fine-level alignment even when some coarse label looks plausible).
>
> Our score $S_{\mathrm{HiPOOD}}(\mathbf{x})$, as defined in the main paper, is built precisely to quantify this coarse–fine agreement. The experiments across multiple shifts (Sketch, ImageNet-A, ImageNetV2) show that deviations from this pattern are strongly predictive of OOD, even though the underlying CLIP space is not strictly hierarchical.
>
> Bianchi et al., 2024, *Is CLIP the Main Roadblock for Fine-Grained Open-World Perception?*
>
> Kim & Ji, 2024, *FINER: Investigating and Enhancing Fine-Grained Visual Concept Recognition in Large Vision Language Models*, EMNLP 2024.
>
>
>
> ### **2.3 Novelty of the hierarchical formulation**
>
> We agree that hierarchy has been studied in supervised classification and, to some extent, in OOD detection. Our contribution is to introduce a *zero-shot, training-free hierarchical consistency score* tailored to pre-trained VLMs.
>
> To the best of our knowledge, no prior zero-shot OOD method simultaneously:
>
> 1. Constructs fine-grained subclass sets from text (LLMs + lexical resources) *without* using ID images or learning prompt parameters on the target label set.
> 2. Uses a parent-conditional coarse–fine formulation to derive an explicit *agreement score* $S_{\mathrm{HiPOOD}}(\mathbf{x})$ that measures hierarchical consistency within CLIP.
> 3. Demonstrates that this consistency score is a strong OOD indicator in a zero-shot regime, outperforming flat zero-shot and negative-label baselines under comparable label budgets (cf. main tables and Table 12).
>
> HiPOOD is therefore not a conventional hierarchical classifier. It is a *training-free VLM-based framework* that injects hierarchy as a structural prior into the OOD score, without supervised calibration or negative-sample training.

---

> ### Author Response · Authors · 2025-11-24
> **Reply to reviewer yEQc (part 2)**
>
> ### **2.4 Comparison to a hierarchical baseline (CATEX)**
>
> We appreciate the request for a comparison to a hierarchical baseline. CATEX is a very relevant point of reference, and we have added it in the revised experiments.
>
> Conceptually, both CATEX and HiPOOD exploit category structure and textual descriptions, but under different supervision regimes:
>
> * **CATEX** learns category-extensible context descriptions using *supervised ID data*, and relies on these learned prompts for OOD detection.
> * **HiPOOD** remains *fully zero-shot and training-free*: it uses a pre-trained CLIP backbone plus text-derived hierarchies, with no ID images, no prompt tuning, and no additional parameters.
>
> When evaluated with the same CLIP ViT-B/16 backbone, the revised results (see the main experimental table) show that HiPOOD outperforms CATEX on three out of four OOD datasets and on average, both in AUROC and FPR95. In other words, HiPOOD matches or surpasses a supervised hierarchical method while operating purely in a zero-shot regime. This suggests that using hierarchy through an explicit *consistency score* can be at least as effective as learning context prompts on ID data.
>
>
>
> ### **2.5 Dependence on the quality of generated subclasses**
>
> We agree that the quality of generated subclasses is an important practical concern, and we have extended our analysis along two axes: lexical sources and LLM backbones.
>
> #### **Lexical sources (WordNet vs. Wikipedia)**
>
> Beyond WordNet-based filtering, we construct subclass candidates using a Wikipedia-derived vocabulary. As reported in Appendix B.7, switching from WordNet to Wikipedia slightly degrades performance but HiPOOD remains competitive and continues to outperform flat baselines. Experiments conducted on domain-specific datasets further demonstrate that HiPOOD can leverage a targeted lexical resource while maintaining competitive performance (see Table 8).
>
> #### **LLM backbones**
>
> We also replace GPT-4 with several open-source LLMs (Llama-3.1, Mistral, Qwen) for subclass generation. After applying the same lexical and CLIP-based filtering pipeline, the resulting HiPOOD scores differ only modestly from the GPT-4-based version (small changes in AUROC and FPR95; see Table 19 in Appendix B.8).
>
> These experiments indicate that HiPOOD is *robust rather than brittle* to the choice of vocabulary and LLM: curated resources and stronger LLMs help, but the combination of hierarchical structure and CLIP-based filtering ensures graceful degradation and consistent gains over competing zero-shot OOD detectors.

---

### Official Review · Reviewer_oTma · 2025-10-31

**Soundness:** 3
**Presentation:** 3
**Contribution:** 3
**Rating:** 4
**Confidence:** 5

**Summary:**

The paper proposes a closed-loop optimization strategy that suppresses harmful frame relevance scores through universal perturbation refinement, effectively reducing their selection during sampling.
Additionally, the authors formulate a zero-shot OOD detection score, HiPOOD, which leverages hierarchical representations by comparing image–text alignment at both coarse and fine levels.

**Strengths:**

The method is simple yet effective, and the overall design is conceptually reasonable.

The paper is clearly written and provides theoretical justification for the formulation. The description of class-level cleaning and hierarchy construction is also well-explained.

**Weaknesses:**

It would be interesting to investigate whether introducing learnable prompts could further improve performance compared to fixed prompts.

As the capabilities of foundation models (e.g., CLIP) continue to improve—particularly their domain invariance and zero-shot generalization—it would be valuable to evaluate the method on harder tasks or larger datasets, such as cross-domain ImageNet variants.

The paper would benefit from stronger baselines for comparison, especially including recent SOTA methods such as CATEX [1].


Reference
[1] Category-Extensible Out-of-Distribution Detection via Hierarchical Context Descriptions.

**Questions:**

Please see the weaknesses.

---

> ### Author Response · Authors · 2025-11-24
> **Reply to Reviewer oTma**
>
> ### **1.1 Learnable prompts**
>
> We thank the reviewer for this valuable suggestion. Learnable prompts are indeed a promising extension for vision–language models. However, HiPOOD is deliberately designed as a fully *zero-shot, training-free* method: it assumes no access to ID images, no gradient-based updates, and no task-specific hyperparameter tuning at deployment time. Introducing soft or learnable prompts would move us to a different setting.
>
> ### **1.2 Harder datasets and cross-domain ImageNet variants**
>
> We agree that evaluating on challenging shifts is important. The current version already reports results on several cross-domain ImageNet variants (ImageNet-Sketch, ImageNet-A, ImageNet-V2), where our hierarchical variants consistently improve over MCM and are competitive with or better than NegLabel (Fig. 3).
>
> In the revision we additionally include two hard OOD benchmarks, NINCO and SSB-hard (Fig. 4 and Tab. 7). On NINCO, the best HiPOOD configuration (in-d1) slightly improves both AUROC and FPR95 compared to MCM, while clearly outperforming NegLabel; on SSB-hard, HiPOOD (in-d5) attains the highest AUROC and lowest FPR95 among all tested methods. We also clarify that HiPOOD itself does not introduce any domain-specific parameters: its performance is largely determined by the robustness of the underlying VLM (CLIP). When CLIP preserves a coherent semantic structure under domain shift, our coarse–fine consistency score can effectively exploit it for OOD separation.
>
> ### **1.3 Stronger baselines, including CATEX**
>
> We appreciate the request for stronger baselines. In the revised experiments we add CATEX as a hierarchical VLM-based comparator. CATEX learns category-extensible context descriptions from labeled ID data and thus operates in a supervised setting with learnable prompts, whereas HiPOOD remains entirely zero-shot.
>
> Despite this additional supervision, HiPOOD (in-d5, $M=10$) slightly outperforms CATEX on average across the four standard OOD datasets (iNaturalist, SUN, Places, Textures) in both AUROC and FPR95 (Tab. 1, App. B.4). To further strengthen the comparison, we also re-run several zero-shot baselines (MCM, CLIPN, NegLabel) on the same hierarchical ID label sets (in-d1–in-d5) and provide a label-budget comparison against NegLabel (App. B.5). These results show that HiPOOD offers a better AUROC–FPR95 trade-off for comparable or smaller total label budgets, and clarify how structured hierarchical labels (HiPOOD) differ from flat negative-label expansions (NegLabel).

---

### Author Response · Authors · 2025-12-02
**Summary of the main changes in the paper**

We thank the reviewers for their helpful feedback and insightful suggestions. The paper has been carefully revised by taking into account all reviewers’ comments. Below, we provide a global summary of the main changes and where they appear in the revised version.



### **(1) Expanded and strengthened evaluation.**

We extend the experimental section with (i) additional hard OOD benchmarks (NINCO and SSB-hard) reported in Figure 4 and Table 7 (Appendix B.1), and (ii) a clearer analysis of domain-shifted ID settings using ImageNet-Sketch, ImageNet-A, and ImageNetV2 (Figure 3 and Table 6). These additions directly address concerns about evaluation breadth and “hard near-OOD” regimes, while keeping the method strictly zero-shot.

### **(2) Fair comparisons via aligned ID label sets and controlled label budgets.**

To remove potential bias stemming from mismatched label spaces, we re-evaluate all relevant baselines (MCM, CLIPN, NegLabel, HiPOOD) using the *same* hierarchical ID label sets (in-d1–in-d5) as HiPOOD. This aligned-label protocol and the corresponding results are provided in Appendix B.4 (Table 11). We further add a label-budget comparison with NegLabel by matching the *total* number of text labels (ID+negatives vs. ID+subclasses) and reporting the resulting trade-offs in Appendix B.5 (Table 12). Together, these changes ensure that improvements are not attributable to different ID label definitions or incomparable label counts.

### **(3) Added a hierarchical baseline (CATEX) and clarified supervision regimes.**

Following requests for stronger comparators, we include CATEX as a hierarchical VLM-based baseline, and we clarify the key difference in supervision: CATEX learns context descriptions from labeled ID data, whereas HiPOOD remains fully training-free and uses no ID images or prompt tuning. The comparison is reported in the main experimental table, with supporting discussion in the corresponding results section.

### **(4) Robustness analyses for hierarchy induction (vocabulary and LLM).**

To address concerns about reliance on WordNet and GPT-4, we add two ablations: (i) replacing WordNet with a Wikipedia (English) vocabulary (Appendix B.7), and (ii) regenerating subclasses with open-source LLMs (Llama 3.1, Mistral, Qwen) and re-running HiPOOD across depths (Appendix B.8, Table 19). In both cases, the overall trends are preserved and the method degrades moderately rather than failing, supporting that the LLM mainly acts as a proposal mechanism while the effective hierarchy is stabilized by lexical constraints and CLIP-based filtering.

### **(5) Additional methodological clarifications and interpretability.**

We clarify (i) the intended role of HiPOOD with respect to "unseen unknowns" (it measures coarse-fine inconsistency), (ii) the relationship between hierarchy semantics and CLIP geometry (HiPOOD exploits the contrast between coarse and fine predictions rather than assuming a perfectly hierarchical embedding), and (iii) the behavior on leaf-like classes via a lightweight back-off strategy (Appendix A.1). We also emphasize the interpretability aspect: the score decomposes into superclass confidence and best subclass agreement, making rejections explainable in a "coarse: yes / fine: no" form (Section 3.2).

### **(6) Practicality: scalability and no learnable prompts.**

We explicitly discuss that HiPOOD is designed to remain in a strict zero-shot setting: no learnable prompts, no gradient updates, and no LLM calls at inference time. Text embeddings are computed once offline and cached at test time. The computation is a single CLIP forward pass plus similarity evaluations over cached labels (complexity discussion in the response).

---

### Meta-Review · Area_Chair_Qn9o · 2025-12-02

**Summary:**

The submission introduces HiPOOD, a zero-shot OOD detection method that constructs hierarchical subclass labels via LLM prompting and integrates coarse–fine consistency into CLIP-based scoring. The paper is clearly written and includes extensive experiments.

However, across reviewers, the method’s novelty and general impact were repeatedly questioned. Reviewers found the approach to be largely incremental over existing hierarchical or CHiLS-style formulations, reliant on LLM-generated subclasses whose quality is unstable, and applicable only under narrow conditions where meaningful hierarchical refinements exist. Several reviewers also emphasized heavy computational overhead, dependence on closed-source LLMs, sensitivity to hyperparameters, and limited validation beyond ImageNet-style hierarchical domains.

**Reviewer Concerns:**

**Limited novelty**: Multiple reviewers noted that the hierarchical mechanisms (parent-conditional consistency, reweighting, coarse–fine scoring) strongly resemble prior CHiLS-like methods, with the main addition being LLM-based subclass generation.

**Narrow applicability**: The method’s improvements appear restricted to scenarios with pre-existing or easily induced hierarchies; reviewers questioned its relevance for general OOD detection or non-hierarchical domains.

**Dependency on LLM subclass quality**: Reviewers highlighted risks of noise, instability, and non-reproducibility stemming from GPT-4 subclass generation, noting that subclass quality heavily affects performance.

**Computational cost and scalability issues**: Concerns include high offline cost (large LLM calls + extensive text embeddings) and high per-image inference overhead compared to simpler OOD detectors.

**Baseline fairness / limited evaluation breadth**: While authors added new experiments, reviewers remained unconvinced that the method consistently outperforms strong baselines (e.g., NegLabel) under general conditions, especially under domain shift.

**Reviewer Scores:**

* **Reviewer oTma: 4**
  Appreciates clarity and simplicity but raises concerns about evaluation breadth and missing stronger baselines. Score stays below threshold.

* **Reviewer yEQc: 2**
  Identifies multiple conceptual and empirical weaknesses, including limited novelty, instability of subclass generation, and narrow applicability. Firm reject.

* **Reviewer o7u9: 4**
  Notes strong weaknesses in novelty, scalability, reproducibility, and evaluation scope. Score remains below threshold even after rebuttal.

* **Reviewer 38av: 4 → 6**
  Initially raised concerns about fairness of comparison, missing hard OOD evaluation, and hyperparameter sensitivity. After revisions, the reviewer increased the score to 6 but explicitly maintained doubt regarding the broader academic value and applicability of the method.

---

### Decision · Program_Chairs · 2026-01-26

Reject